# RoPAWS: Robust Semi-supervised Representation Learning from Uncurated Data

**Sangwoo Mo**[1][*]    **Jong-Chyi Su**[2]    **Chih-Yao Ma**[3]    **Mahmoud Assran**[3,4,5]
**Ishan Misra**[3]    **Licheng Yu**[3]    **Sean Bell**[3]
[1]KAIST    [2]NEC Laboratories America    [3]Meta AI    [4]McGill University    [5]Mila

## Abstract

Semi-supervised learning aims to train a model using limited labels. State-of-the-art semi-supervised methods for image classification such as PAWS rely on self-supervised representations learned with large-scale unlabeled but curated data. However, PAWS is often less effective when using real-world unlabeled data that is uncurated, e.g., contains out-of-class data. We propose RoPAWS, a robust extension of PAWS that can work with real-world unlabeled data. We first reinterpret PAWS as a generative classifier that models densities using kernel density estimation. From this probabilistic perspective, we calibrate its prediction based on the densities of labeled and unlabeled data, which leads to a simple closed-form solution from the Bayes' rule. We demonstrate that RoPAWS significantly improves PAWS for uncurated Semi-iNat by +5.3% and curated ImageNet by +0.4%.[1]

## 1 Introduction

Semi-supervised learning aims to address the fundamental challenge of training models with limited labeled data by leveraging large-scale unlabeled data. Recent works exploit the success of self-supervised learning (He et al., 2020; Chen et al., 2020a) in learning representations from unlabeled data for training large-scale semi-supervised models (Chen et al., 2020b; Cai et al., 2022). Instead of self-supervised pre-training followed by semi-supervised fine-tuning, PAWS (Assran et al., 2021) proposed a single-stage approach that combines supervised and self-supervised learning and achieves state-of-the-art accuracy and convergence speed.

While PAWS can leverage curated unlabeled data, we empirically show that it is not robust to real-world uncurated data, which often contains out-of-class data. A common approach to tackle uncurated data in semi-supervised learning is to filter unlabeled data using out-of-distribution (OOD) classification (Chen et al., 2020d; Saito et al., 2021; Liu et al., 2022). However, OOD filtering methods did not fully utilize OOD data, which could be beneficial to learn the representations especially on large-scale realistic datasets. Furthermore, filtering OOD data could be ineffective since in-class and out-of-class data are often hard to discriminate in practical scenarios.

To this end, we propose RoPAWS, a robust semi-supervised learning method that can leverage uncurated unlabeled data. PAWS predicts out-of-class data overconfidently in the known classes since it assigns the pseudo-label to nearby labeled data. To handle this, RoPAWS regularizes the pseudo-labels by measuring the similarities between labeled and unlabeled data. These pseudo-labels are further calibrated by label propagation between unlabeled data. Figure 1 shows the conceptual illustration of RoPAWS and Figure 4 visualizes the learned representations.

More specifically, RoPAWS calibrates the prediction of PAWS from a probabilistic view. We first introduce a new interpretation of PAWS as a generative classifier, modeling densities over representation by kernel density estimation (KDE) (Rosenblatt, 1956). The calibrated prediction is given by a closed-form solution from Bayes' rule, which implicitly computes the fixed point of an iterative propagation formula of labels and priors of unlabeled data. In addition, RoPAWS explicitly controls out-of-class data by modeling a prior distribution and computing a reweighted loss, making the model robust to uncurated data. Unlike OOD filtering methods, RoPAWS leverages all of the unlabeled (and labeled) data for representation learning.

---

[*]Work done during an internship at Meta AI. Correspondence to: swmo@kaist.ac.kr
[1]Code: https://github.com/facebookresearch/suncet

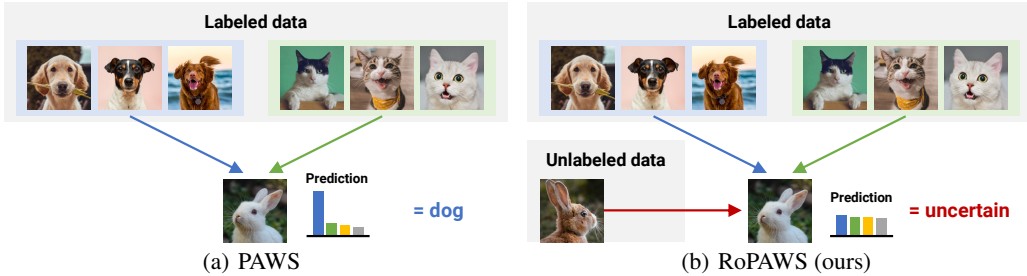

Figure 1: Conceptual illustration of the proposed RoPAWS. PAWS assigns the pseudo-label of unlabeled data by the nearby labeled data; however, this makes the prediction of out-of-class data overconfident. In uncurated setting, unlabeled data contains out-of-class data, for which the model should have uncertain (not confident) predictions. Therefore, RoPAWS regularizes the pseudo-labels by comparing the similarities between unlabeled data and labeled data.

We demonstrate that RoPAWS outperforms PAWS and previous state-of-the-art (SOTA) robust semi-supervised learning methods in various scenarios. RoPAWS improves PAWS in uncurated data with a large margin. In addition, PAWS improves PAWS in curated data by calibrating the prediction of uncertain data. Figure 2 highlights our main results: RoPAWS improves (a) PAWS by +5.3% and previous SOTA by +1.7% on uncurated Semi-iNat (Su & Maji, 2021b), (b) PAWS (previous SOTA) by +0.4% on curated ImageNet (Deng et al., 2009) using 1% of labels, and (c,d) PAWS by +6.9% and +22.7% on uncurated CIFAR-10 (Krizhevsky et al., 2009) using 25 labels for each class, using CIFAR-100 and Tiny-ImageNet as an extra unlabeled data, respectively.

## 2 RELATED WORK

Semi-supervised learning has been an active area of research for decades (Chapelle et al., 2006; Van Engelen & Hoos, 2020). Classic works directly regularize the classifier's prediction, which we call the *prediction-based* approach. Recent works focus more on the representation that stems from the final classifier, which we call the *representation-based* approach. Leveraging the recent progress of self-supervised learning (He et al., 2020), the representation-based approach has more promise in large-scale scenarios and can be more robust under uncurated data. In the following subsections, we will briefly review the general idea of each approach and how they handle uncurated data.[2]

### 2.1 PREDICTION-BASED SEMI-SUPERVISED LEARNING

**General approach.** Prediction-based approaches regularize the classifier's prediction of unlabeled data. Two objectives are popularly used: pseudo-labeling and consistency regularization. Pseudo-labeling (Lee et al., 2013; Yalniz et al., 2019; Xie et al., 2020b; Rizve et al., 2021; Cascante-Bonilla et al., 2021; Hu et al., 2021) predicts labels of unlabeled data and retrains the classifier using the predicted labels, minimizing the prediction entropy of unlabeled data (Grandvalet & Bengio, 2004). Consistency regularization (Sajjadi et al., 2016; Laine & Aila, 2017; Tarvainen & Valpola, 2017; Miyato et al., 2018; Xie et al., 2020a) enforces the prediction that two views of the same image are similar. Combining two objectives, prediction-based approaches has shown notable results (Berthelot et al., 2019; 2020; Sohn et al., 2020; Kuo et al., 2020; Li et al., 2021). However, they underperform than representation-based approaches for large-scale scenarios (Chen et al., 2020b). Moreover, most prior works assume that unlabeled data are curated, i.e., follow the same distribution of labeled data, and often fail when unlabeled data are uncurated (Oliver et al., 2018; Su et al., 2021).

**Handling uncurated data.** Numerous works have attempted to make the prediction-based approach robust to uncurated data. Most prior works assume the unlabeled data are composed of in- and out-of-domain (OOD) data and filter OOD data by training an OOD classifier (Chen et al., 2020d; Guo et al., 2020; Yu et al., 2020; Huang et al., 2021a;b; Saito et al., 2021; Killamsetty et al., 2021; Nair et al., 2019; Augustin & Hein, 2020; Park et al., 2021). From the perspective of the prediction-based approach, it is natural to filter OOD (particularly out-of-class) data since they are irrelevant to in-

---

[2]Besides the deep learning approaches, SSKDE (Wang et al., 2009) proposed semi-supervised kernel density estimation, which technically resembles RoPAWS. The detailed discussion and comparison are in Appendix H.

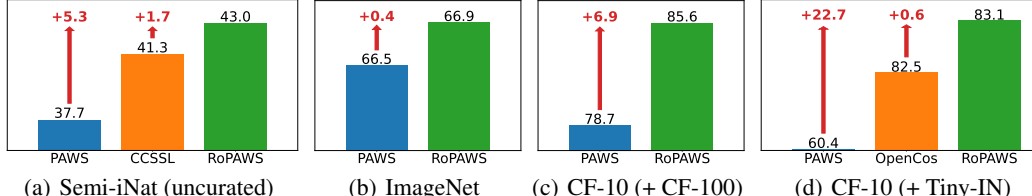

Figure 2: Test accuracy (%) of PAWS, previous SOTA, and RoPAWS on various datasets. CF stands for CIFAR and Tiny-IN stands for Tiny-ImageNet. We use 1% of labels for ImageNet and 25 labels for each class for CIFAR. RoPAWS highly improves PAWS and outperforms previous SOTA.

class prediction. However, the filtering approach has two limitations: (a) it ignores the representation learned from OOD data, and (b) the detection of OOD can be artificial. Note that the model cannot distinguish near OOD and in-class data not covered by labels. Therefore, we do not filter the OOD samples but (a) learn representation from them and (b) consider soft confidence of being in-domain. Some recent works reveal that utilizing OOD data can be better than filtering them (Luo et al., 2021; Han et al., 2021; Huang et al., 2022). RoPAWS is also in this line of work; however, leveraging OOD data is more natural from the representation learning perspective.

## 2.2 REPRESENTATION-BASED SEMI-SUPERVISED LEARNING

**General approach.** Self-supervised learning has shown remarkable success recently, learning a universal representation that can be transferred to various downstream tasks. Specifically, the siamese network approach, which makes the representation of two views from the same image invariant, has dramatically improved image classification (He et al., 2020; Chen et al., 2020a). Recent works reveal that fine-tuning (or training a lightweight classifier on top of) the self-supervised model performs well for semi-supervised learning, especially for high-resolution and large-scale datasets (Zhai et al., 2019; Rebuffi et al., 2020; Chen et al., 2020b; Cai et al., 2022). However, self-supervised pretraining is inefficient when the target downstream task is given; one can directly train a task-specific model. From this motivation, PAWS (Assran et al., 2021) proposed a semi-supervised extension of the siamese network approach, enforcing the prediction (computed from representation) of two views be invariant. Notably, PAWS has shown to be more effective (i.e., performs better) yet efficient (i.e., converges faster) than the two-stage approach of self-supervised pre-training and then fine-tuning. However, PAWS is often not robust to uncurated data, limiting its applicability to real-world scenarios. Thus, we propose RoPAWS, a robust extension of PAWS for uncurated data.

**Handling uncurated data.** Self-supervised learning is robust to uncurated data, learning a generic representation of data, and does not suffer from the issues of labels such as out-of-class, long-tail, or noisy labels (Kang et al., 2020; Tian et al., 2021; Goyal et al., 2021; 2022). Thus, fine-tuning a self-supervised model is a strong baseline for robust semi-supervised learning. However, we claim that semi-supervised learning can maintain its efficacy (and efficiency) while enjoying the robustness of self-supervised learning by designing a robust semi-supervised method called RoPAWS. On the other hand, some works combine self-supervised and supervised learning, where the supervised loss is only applied to the labeled (or confident) data (Yang et al., 2022; Khosla et al., 2020). It is safe but too conservative as it does not propagate labels over unlabeled data. This approach is effective for a related problem called open-world semi-supervised learning, which aims to cluster novel classes while predicting known classes (Cao et al., 2022; Vaze et al., 2022; Rizve et al., 2022; Sun & Li, 2022). In contrast, we focus on predicting the known classes, i.e., no extra burden for novel classes. Extending our work to open-world setups would be an interesting future direction.

## 3 RoPAWS: ROBUST PAWS FOR UNCURATED DATA

We propose RoPAWS, a robust semi-supervised learning method for uncurated data. In the following subsections, we first explain the problem of PAWS (Assran et al., 2021) on uncurated data, then introduce a novel probabilistic interpretation of PAWS. Finally, we describe our proposed RoPAWS, which calibrates the prediction of targets from the probabilistic view for robust training. Figure 3 visualizes the overall framework of RoPAWS. Detailed derivations are provided in Appendix A.

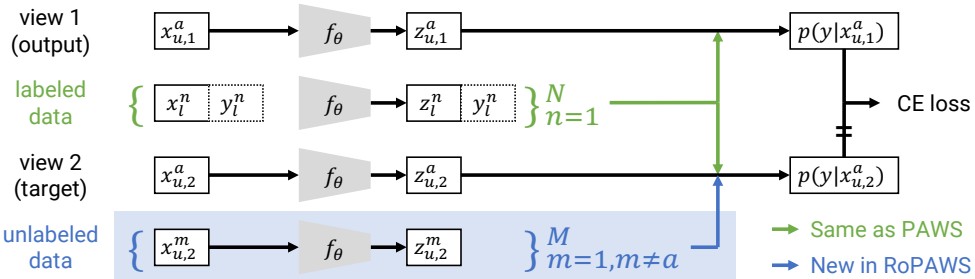

Figure 3: Overall framework for RoPAWS. RoPAWS computes the cross-entropy (CE) loss between the predictions of output and target views. Output prediction comes from the similarities to labeled data, same as PAWS. However, target prediction comes from both labeled and unlabeled data, which calibrates the prediction for uncertain data and makes training robust to uncurated data.

## 3.1 PROBLEM SETUP AND PRELIMINARIES

**Problem setup.** Semi-supervised learning aims to learn a classifier from a large unlabeled dataset $\mathcal{D}_u := \{x_u\}$ and a small labeled dataset $\mathcal{D}_l := \{x_l, y_l\}$ for $\{x_l\} \subset \mathcal{D}_u$. Most prior works assume that $\mathcal{D}_l$ and $\mathcal{D}_u$ are drawn from the same distribution. However, the unlabeled data are often *uncurated*, i.e., containing uncertain (far from labels, often distribution-shifted) or out-of-class data. We aim to extend the applicability of semi-supervised learning for these uncurated scenarios.

**Preliminaries.** In this paper, we focus on semi-supervised for image classification. We start with introducing PAWS (Assran et al., 2021), which is the state-of-the-art method. PAWS leverages the power of self-supervised learning, particularly the siamese network approach (Caron et al., 2020), to learn useful representation from unlabeled data. Specifically, PAWS enforces the predictions of two views from the same image to be invariant. Unlike self-supervised learning, PAWS defines the prediction using labeled samples. Formally, let $\mathcal{B}_l = \{x_l^i, y_l^i\}_{i=1}^N$ and $\mathcal{B}_u = \{x_u^i\}_{i=1}^M$ be the labeled and unlabeled batches of size $N$ and $M$, sampled from $\mathcal{D}_l$ and $\mathcal{D}_u$, respectively. Here, the labeled batch $\mathcal{B}_l$ is class-balanced, i.e., $|\mathcal{B}_l^y| = \frac{N}{C}$ for $\mathcal{B}_l^y := \{(x_l^i, y_l^i) \in \mathcal{B}_l \mid y_l^i = y\}$ for all the classes $y \in [C] := \{1, .., C\}$. PAWS samples a subset of classes for each mini-batch.

Given the batches, PAWS first computes normalized embedding $z = f_\theta(x)/\|f_\theta(x)\|$, i.e., $\|z\| = 1$ using an encoder $f_\theta(\cdot)$. For simplicity, we omit the encoder $f_\theta(\cdot)$ in the future equations and denote the normalized embedding by substituting $x$ with $z$ from the original data $x$, i.e., $f_\theta(x_{\text{sub}}^{\text{super}}) = z_{\text{sub}}^{\text{super}}$. We assume that an embedding $z$ is a $1 \times d$ vector, and $\mathbf{z}_l$ and $\mathbf{z}_u$ are $N \times d$ and $M \times d$ matrices represent the embeddings of labeled and unlabeled data. Here, PAWS predicts the label $p(y|x)$ of data $x$ by finding the nearest labeled sample $x_l$ and softly assigning the label $y_l$. Note that we compute the similarity on the representation space and $p(y|x)$ is a function of $z$. Then,

$$p_{\text{PAWS}}(y|x) := \sigma_\tau(z \cdot \mathbf{z}_l^\mathsf{T}) \cdot \mathbf{p}_l^{y|x} \tag{1}$$

where $\mathbf{p}_l^{y|x} \in \mathbb{R}^{N \times C}$ are the corresponding one-hot (or soft) labels with a row-wise sum of 1, and $\sigma_\tau(\cdot)$ is the softmax function with temperature $\tau$. PAWS then enforces the prediction of two views $p_1^i := p(y|t_1(x^i))$ and $p_2^i := p(y|t_2(x^i))$ from the same unlabeled data $x^i$ with augmentations $t_1, t_2 \sim \mathcal{T}$ to be identical. Additionally, the sharpening function $\rho(\cdot)$ is applied to targets which are detached from backpropagation. PAWS further regularizes the average sharpened prediction $\bar{p} := \frac{1}{2M} \sum_i (\rho(p_1^i) + \rho(p_2^i))$ to be uniform. To sum up, the training objective of PAWS is:

$$\mathcal{L}_{\text{PAWS}} := \frac{1}{2M} \sum_{x_u^i \in \mathcal{B}_u} \left( \mathbb{H}(p_1^i, \rho(p_2^i)) + \mathbb{H}(p_2^i, \rho(p_1^i)) \right) - \mathbb{H}(\bar{p}) \tag{2}$$

where $\mathbb{H}(\cdot, \cdot)$ is cross-entropy and and $\mathbb{H}(\cdot)$ is entropy. Recall that the prediction is computed from the labeled data, and thus the objective is a function of both $\mathcal{B}_l$ and $\mathcal{B}_u$.

**Problem with PAWS.** The prediction of PAWS is overconfident to given $[C]$ classes since it assigns the label of the nearby labeled data. However, it can be problematic for uncurated data containing out-of-class data. Specifically, PAWS equally assigns high confidence for both in- and out-of-class data, unlike RoPAWS assigns high confidence for in-class and low confidence for out-of-class data (see Appendix J). Thus, we aim to calibrate the prediction to make PAWS robust.

## 3.2 PROBABILISTIC VIEW OF PAWS

Our key observation is that PAWS can be interpreted as a generative classifier, which estimates the densities over the representation space. To this end, we consider the kernel density estimation (KDE) (Rosenblatt, 1956) using the kernel as the similarity on the representation space:[3]

$$p(x) := \frac{1}{|\mathcal{D}|} \sum_{x' \in \mathcal{D}} k(z, z') \quad \text{for} \quad k(z, z') := \frac{1}{Z} \cdot \exp(z \cdot z'/\tau) \tag{3}$$

where $Z$ is a normalizing constant to make $\frac{1}{Z} \int_{z'} k(z, z') = 1$. From the definition, we can estimate the conditional $p(x|y)$ and marginal $p(x)$ densities using the labeled batch of each class $\mathcal{B}_l^y$ and the entire labeled batch $\mathcal{B}_l$, respectively. Formally, the densities of PAWS are:

$$p_{\text{PAWS}}(x|y) := \frac{1}{|\mathcal{B}_l^y|} \sum_{x' \in \mathcal{B}_l^y} k(z, z') \quad \text{and} \quad p_{\text{PAWS}}(x) := \frac{1}{|\mathcal{B}_l|} \sum_{x' \in \mathcal{B}_l} k(z, z'). \tag{4}$$

Applying them to the Bayes' rule, we can compute the prediction $p(y|x)$:[4]

$$p_{\text{PAWS}}(y|x) = \frac{p_{\text{PAWS}}(x|y)\, p(y)}{p_{\text{PAWS}}(x)} = \frac{\sum_{\mathcal{B}_l^y} k(z, z')}{\sum_{\mathcal{B}_l} k(z, z')} = \sum_{\mathcal{B}_l^y} \frac{\exp(z \cdot z'/\tau)}{\sum_{\mathcal{B}_l} \exp(z \cdot z'/\tau)} = \sigma_\tau(z \cdot \mathbf{z}_l^\mathsf{T}) \cdot \mathbf{p}_l^{y|x} \tag{5}$$

where $\mathbf{p}_l^y$ are one-hot labels. Remark that it recovers the original prediction of PAWS in Eq. (1).

We further extend the densities to consider soft labels, e.g., PAWS applies label smoothing (Müller et al., 2019), using the weighted kernel $k^y(z, z') := \frac{1}{K} p(y|x') k(z, z')$:

$$p_{\text{PAWS-soft}}(x|y) = \frac{1}{NK} \sum_{x' \in \mathcal{B}_l} p(y|x')\, k(z, z') \quad \text{and} \quad p_{\text{PAWS-soft}}(x) = \frac{1}{NK} \sum_{x' \in \mathcal{B}_l} \frac{1}{C} k(z, z') \tag{6}$$

assuming that the normalizing constant $K$ is identical for all classes (class-balanced). Then, this definition also recovers the original prediction of PAWS considering soft labels:

$$p_{\text{PAWS-soft}}(y|x) = \sum_{\mathcal{B}_l} \frac{\exp(z \cdot z'/\tau)}{\sum_{\mathcal{B}_l} \exp(z \cdot z'/\tau)} \cdot p(y|x') = \sigma_\tau(z \cdot \mathbf{z}_l^\mathsf{T}) \cdot \mathbf{p}_l^{y|x}. \tag{7}$$

To summarize, PAWS is a generative classifier, and one can estimate the density of a sample on the representation space learned by PAWS. This probabilistic view gives us a principled way to extend PAWS for various scenarios, e.g., robust training on uncurated data, as stated below.

## 3.3 ROBUST PAWS VIA CALIBRATED PREDICTION

Recall that the prediction of PAWS is overconfident to the known classes $[C]$ since it propagates labels from nearby labeled data. Assuming that we also know the labels of unlabeled data, which have less confident values for uncertain (or out-of-class) data, we can calibrate the prediction by considering *both* labels of nearby labeled and unlabeled samples. However, we do not know the labels of unlabeled data, and using them for prediction raises a chicken-and-egg problem.

Interestingly, the probabilistic view gives the solution: from the Bayes' rule, the prediction $p_u(y|x)$ of unlabeled data is a function of densities $p_u(x|y)$ and $p_u(x)$, which are again a function of labels $p_l(y|x)$ of labeled data and predictions $p_u(y|x)$ of unlabeled data. Solving the equation, we can get a closed-form solution of $p_u(y|x)$ as a function of labels $p_l(y|x)$ and distances of embeddings $\mathbf{z}_l, \mathbf{z}_u$. To this end, we first extend the densities in Eq. (6) to consider both labeled and unlabeled data:[5]

$$p_{\text{RoPAWS}}(x|y) = \frac{1}{K'} \sum_{\mathcal{B}_l \cup \mathcal{B}_u} p(y|x')\, k(z, z') \quad \text{and} \quad p_{\text{RoPAWS}}(x) = \frac{1}{K'} \sum_{\mathcal{B}_l \cup \mathcal{B}_u} \frac{1}{C} k(z, z'). \tag{8}$$

---

[3] Lee et al. (2018b) suggested a generative classifier view of deep representations using the Gaussian mixture model (GMM). However, GMM needs enough labeled samples, and KDE fits more to a few label setting.

[4] Recall that the labeled batch is class-balanced, hence $|\mathcal{B}_l| = N$, $|\mathcal{B}_l^y| = \frac{N}{C}$, and $p(y) = \frac{1}{C}$ for all $y$.

[5] We balance the scale of labeled and unlabeled data with a ratio $r : 1$ when the unlabeled batch is large.

where $K' = (N + M)K$ is the normalizing constant. Applying them to the Bayes' rule:[6]

$$p_{\text{RoPAWS}}(y|x) = \sigma_\tau(z \cdot [\mathbf{z}_l^\mathsf{T} \mid \mathbf{z}_u^\mathsf{T}]) \cdot [\mathbf{p}_l^{y|x} \mid \mathbf{p}_u^{y|x}]^\mathsf{T} \cdot C \, p(y)$$
$$= s_l \cdot \mathbf{p}_l^{y|x} \cdot C \, p(y) + s_u \cdot \mathbf{p}_u^{y|x} \cdot C \, p(y) \tag{9}$$

where $[s_l | s_u] := \sigma_\tau(z \cdot [\mathbf{z}_l^\mathsf{T} | \mathbf{z}_u^\mathsf{T}]) \in \mathbb{R}^{1 \times (N+M)}$ are similarities of a sample $x$ to labeled and unlabeled data. Aggregating the predictions of unlabeled data, we can formulate the equation in a matrix form:

$$\mathbf{p}_u^{y|x} = (\mathbf{s}_l \odot C \, \mathbf{p}_u^y) \cdot \mathbf{p}_l^{y|x} + (\mathbf{s}_u \odot C \, \mathbf{p}_u^y) \cdot \mathbf{p}_u^{y|x} \tag{10}$$

where $\mathbf{s}_l, \mathbf{s}_u$ are matrix versions of $s_l, s_u$, $\mathbf{p}_u^y$ is a matrix version of $p(y)$, and $\odot$ is an element-wise product. By organizing the formula, we get the prediction $q(y|x)$ of RoPAWS:[7]

$$\mathbf{q}_u^{y|x} := \mathbf{p}_u^{y|x} = (\mathbf{I} - \mathbf{s}_u \odot C \, \mathbf{p}_u^y)^{-1} \cdot (\mathbf{s}_l \odot C \, \mathbf{p}_u^y) \cdot \mathbf{p}_l^y \in \mathbb{R}^{M \times C}. \tag{11}$$

This formula can also be interpreted as label propagation (Zhou et al., 2003; Bengio et al., 2006; Iscen et al., 2019) of labels $\mathbf{p}_l^{y|x}$ and priors $\mathbf{p}_u^y$ of unlabeled data to get the posterior $\mathbf{p}_u^{y|x}$.

**In-domain prior.** Considering both nearby labeled and unlabeled samples gives a better calibrated prediction but has a threat of amplifying the wrong prediction of unlabeled data. Indeed, propagating the labels from (certain) labels and (uncertain) predictions of unlabeled data equally can be too risky. Thus, we further improve the prediction formula by putting more weights on certain samples.

This modification can be naturally done under the proposed framework by controlling the prior $p(y)$ to consider out-of-class data. Specifically, we give a data-dependent prior of being in-domain (i.e., close to labeled data) for unlabeled data. Let $\mathbf{p}_u^{\text{in}} \in \mathbb{R}^{M \times 1}$ be the in-domain prior for each sample. Then, we set the prior belief $p(y)$ of in-class as $\mathbf{p}_u^{\text{in}}/C$ (class-balanced). Here, we only adopt the prior belief of unlabeled data, and the densities $p(x|y)$ and $p(x)$ remains the same as in Eq. (8). Thus, applying the adopted belief of the prior gives the regularized prediction formula:

$$\mathbf{q}_u^{y|x} = (\mathbf{I} - \mathbf{s}_u \odot \mathbf{p}_u^{\text{in}})^{-1} \cdot (\mathbf{s}_l \odot \mathbf{p}_u^{\text{in}}) \cdot \mathbf{p}_l^y. \tag{12}$$

The final remaining step is to estimate the in-domain prior $\mathbf{p}_u^{\text{in}}$. Here, we assume that a data is likely to be in-domain if it is close to some labeled data. To this end, we measure the similarity between data $x$ and labeled data $\{x_l^i\}$ using the kernel defined in Eq. (3). However, we normalize the kernel to make the similarity between $x_l^i$ and $x_l^i$ itself to be 1 (i.e., in-domain prior of $x_l^i$ is 1) and use the different temperature $\tau_{\text{prior}}$ which controls the overall confidence of in-domain prior; use lower $\tau_{\text{prior}}$ for when data are more uncertain. Putting it all together, the in-domain prior is given by:

$$p(\text{in} \mid x) := \max_{x_l^i \in \mathcal{B}_l} \, \exp\left(\frac{z \cdot z_l^i - 1}{\tau_{\text{prior}}}\right) \in (0, 1] \tag{13}$$

where $\mathbf{p}_u^{\text{in}}$ is a matrix version of $p(\text{in}|x)$. Note that $p(\text{in}|x_l^i) = \exp(0) = 1$.

We remark that setting $p(\text{in}|x) < 1$ implicitly assigns the remaining probability to the background class. However, unlike the background class enforcing the uncertain samples to be collapsed into a few regions (Dhamija et al., 2018), RoPAWS implicitly considers them and all computations are done in the label space of $[C]$. Here, the prediction $q(y|x) \in \mathbb{R}^C$ from Eq. (12) has the sum $< 1$, where this sum gives a posterior probability of being in-domain $q(\text{in}|x)$. We convert this prediction $q(y|x)$ to the normalized $C$-class probability by adding a uniform multiplied by $1 - q(\text{in}|x)$, i.e., uncertain samples are close to uniform (Lee et al., 2018a), gives the final prediction values.

**Training objective.** Recall that the training objective of PAWS is to enforce the prediction of view 1 (output) to be identical to the prediction of view 2 (target). Here, we apply the calibrated prediction in Eq. (12) only to the target (denoted as $q^i$) and keep the original prediction in Eq. (1) for the output (denoted as $p^i$). We use the same inference (soft nearest-neighbor or fine-tuning) with PAWS, meaning that we keep the output the same but only calibrate the target.

RoPAWS regularizes the prediction of uncertain (or out-of-class) data to be uniform, which may make the prediction too unconfident. To avoid this, we give less weight to the objectives of uncertain

---

[6]We do not cancel out $C$ and $p(y)$ since we will control the prior $p(y)$ later, to consider out-of-class data.

[7]We denote the posterior prediction of RoPAWS as $q(y|x)$ to discriminate with PAWS later.

Table 1: Test accuracy (%) of Semi-SL methods trained on Semi-iNat using ResNet-50, using either a curated or uncurated (superset of curated) unlabeled data. Subscripts denote the gain over PAWS. RoPAWS improves PAWS for all scenarios, and shows that larger uncurated data outperform curated data. Also, RoPAWS outperforms previous state-of-the-art methods initialized from MoCo-v2.

| Method | From scratch | | ImageNet pre-trained | |
|---|---|---|---|---|
| | Curated | Uncurated | Curated | Uncurated |
| Pseudo-label | 18.6 | 18.8 | 40.3 | 40.3 |
| FixMatch | 15.5 | 11.0 | 44.1 | 38.5 |
| Self-training | 20.3 | 19.7 | 42.4 | 41.5 |
| FixMatch + CCSSL | - | 31.2 | - | - |
| MoCo-v2 + Fine-tuning | 30.2 | 31.8 | 41.7 | 40.8 |
| MoCo-v2 + Self-training | 32.0 | 32.9 | 42.6 | 41.5 |
| MoCo-v2 + FixMatch + CCSSL | - | 41.3 | - | - |
| PAWS | 39.2 | 37.7 | 49.7 | 43.0 |
| RoPAWS (ours) | 39.4 (+0.2) | 43.0 (+5.3) | 50.3 (+0.6) | 44.3 (+1.3) |

Table 2: Test accuracy (%) of Semi-SL methods trained on ImageNet using ResNet-50. We compare prediction-based Semi-SL, Self-SL pre-training and fine-tuning, and representation-based Semi-SL approaches. Subscripts denote the gain over PAWS. RoPAWS achieves state-of-the-art.

| | Semi-SL (pred-based) | | | Self-SL + Fine-tune | | | Semi-SL (rep-based) | |
|---|---|---|---|---|---|---|---|---|
| | UDA | FixMatch | MPL | BYOL | SwAV | SimCLRv2 | PAWS | RoPAWS (ours) |
| Label 1% | - | - | - | 53.2 | 53.9 | 57.9 | 66.5 | 66.9 (+0.4) |
| Label 10% | 68.1 | 71.5 | 73.9 | 68.8 | 70.2 | 68.4 | 75.5 | 75.7 (+0.2) |

samples (Ren et al., 2020). We simply set the weight $w^i$ as the average in-domain posterior of two views $w^i := (\frac{1}{2}(q(y|x_1^i) + q(y|x_2^i)))^k$, where $k$ controls the power of reweighting.

To sum up, the objective of RoPAWS is (differences from PAWS in Eq. (2) are highlighted in red):[8]

$$\mathcal{L}_{\text{RoPAWS}} := \frac{1}{2M} \sum_{x_u^i \in \mathcal{B}_u} w^i \left( \mathbb{H}(p_1^i, \rho(q_2^i)) + \mathbb{H}(p_2^i, \rho(q_1^i)) \right) - \mathbb{H}(\bar{p}). \tag{14}$$

## 4    EXPERIMENTS

**Setup.** We follow the default setup of PAWS: ImageNet (Deng et al., 2009) setup on ResNet-50 (He et al., 2016) for large-scale experiments, and CIFAR-10 (Krizhevsky et al., 2009) setup on WRN-28-2 (Zagoruyko & Komodakis, 2016) for small-scale experiments. RoPAWS has three hyperparameters: the scale ratio of labeled and unlabeled batch $r$, the temperature for in-domain prior $\tau_{\text{prior}}$, and the power of reweighted loss $k$. We set $r = 5$ for all experiments, and set $\tau_{\text{prior}} = 3.0$ for ResNet-50 and $\tau_{\text{prior}} = 0.1$ for WRN-28-2. We set $k = 1$ for all experiments except $k = 3$ for fine-tuning from ImageNet pre-trained models. To evaluate, we use the soft nearest-neighbor classifier for Semi-iNat and CIFAR-10 and accuracy after fine-tuning for ImageNet. All the reported results use the same architecture, ResNet-50 for large-scale and WRN-28-2 for small-scale experiments. See Appendix C for additional details and discussions on the choice of hyperparameters.

### 4.1    LARGE-SCALE EXPERIMENTS

**Semi-iNat.** Semi-iNat (Su & Maji, 2021b) is a realistic large-scale benchmark for semi-supervised learning, containing a labeled dataset of 9,721 images with 810 classes, along with two unlabeled datasets: (a) a small curated one of 91,336 images with the same 810 classes and (b) a large uncurated one of 313,248 images with 2,439 classes. We compare RoPAWS (and PAWS) with various baselines: Pseudo-label (Lee et al., 2013), FixMatch (Sohn et al., 2020), Self-training (Hinton et al.,

---

[8]RoPAWS does not suffer from representation collapse similar to PAWS, as discussed in Appendix B.

Table 3: Test accuracy (%) of Semi-SL methods trained on CIFAR-10 using WRN-28-2, using an extra unlabeled data denoted by (+ extra), using (a) no extra data, (b) CIFAR-100, or (c) Tiny-ImageNet as the extra data. Subscripts denote the gain over PAWS. RoPAWS improves PAWS for all scenarios and is comparable with the previous state-of-the-art robust Semi-SL methods.

| | (a) CIFAR-10 (curated, no extra data) | | (b) CIFAR-10 (+ CIFAR-100) | |
|---|---|---|---|---|
| | 25 labels | 400 labels | 25 labels | 400 labels |
| FixMatch | $94.9_{\pm 0.7}$ | $95.7_{\pm 0.1}$ | 78.1 | 94.1 |
| OpenMatch | 51.6 | 95.1 | 41.9 | 94.1 |
| PAWS | $92.4_{\pm 2.1}$ | $95.1_{\pm 0.5}$ | $78.7_{\pm 2.9}$ | $91.2_{\pm 0.3}$ |
| RoPAWS (ours) | $94.7_{\pm 0.3}$ (+2.3) | $95.7_{\pm 0.2}$ (+0.6) | $85.6_{\pm 3.0}$ (+6.9) | $93.9_{\pm 0.1}$ (+2.7) |

(c) CIFAR-10 (+ Tiny-ImageNet)

| | SimCLR pre-trained | | | | | From scratch | |
|---|---|---|---|---|---|---|---|
| | FixMatch | UASD | DS$^3$L | OpenMatch | OpenCos | PAWS | RoPAWS (ours) |
| 25 labels | $70.5_{\pm 1.2}$ | $74.0_{\pm 0.4}$ | $69.1_{\pm 2.3}$ | $77.9_{\pm 0.8}$ | $82.5_{\pm 1.3}$ | $60.4_{\pm 5.3}$ | $83.1_{\pm 1.8}$ (+22.7) |

2014), and CCSSL (Yang et al., 2022), trained from scratch or pre-trained MoCo-v2 (Chen et al., 2020c). We include the results reported in CCSSL and Semi-iNat (Su & Maji, 2021a).

Table 1 presents the results on Semi-iNat. RoPAWS achieves 43.0% of accuracy for the uncurated, from scratch setup, giving +5.3% of gain over PAWS and improving the state-of-the-art by +1.7%. Note that the prior art, CCSSL requires a pre-training of MoCo-v2 and takes a longer training time. Also, remark that RoPAWS trained on uncurated data from scratch exceeds the one trained on curated data, verifying the benefit of representation learned from large uncurated data. In addition, RoPAWS improves the curated setups since it regularizes the prediction of uncertain in-class samples.[9] Finally, RoPAWS further boosts the accuracy when using an ImageNet pre-trained model.[10] Note that learning representations for the downstream task is still important when fine-tuning a model (Reed et al., 2022), and RoPAWS gives an effective solution for this practical scenario.

**ImageNet.** Following prior works, we provide (curated) ImageNet (Deng et al., 2009) experiments using the full ImageNet as unlabeled data and its 1% or 10% subset as labeled data. We compare RoPAWS and PAWS (state-of-the-art) with various baselines: (a) prediction-based semi-supervised learning (Semi-SL) such as UDA (Xie et al., 2020a), FixMatch (Sohn et al., 2020), and MPL (Pham et al., 2021), and (b) pre-training with self-supervised learning (Self-SL) and fine-tuning such as BYOL (Grill et al., 2020), SwAV (Caron et al., 2020), and SimCLRv2 (Chen et al., 2020b). PAWS and RoPAWS fall into the representation-based Semi-SL. Table 2 presents RoPAWS outperforms PAWS and achieves state-of-the-art, e.g., +0.4% for 1% labels.

### 4.2 SMALL-SCALE EXPERIMENTS

**CIFAR-10.** We consider three settings for CIFAR-10: using (a) no extra data (curated setting), (b) CIFAR-100 (hard OOD), and (c) Tiny-ImageNet (easy OOD), as extra unlabeled data. We compare RoPAWS with PAWS and FixMatch (Sohn et al., 2020), together with state-of-the-art robust Semi-SL (OOD filtering) methods: UASD (Chen et al., 2020d), DS$^3$L (Guo et al., 2020), Open-Match (Saito et al., 2021), and OpenCos (Park et al., 2021). We use the reported results from the FixMatch paper for setup (a) and the OpenCos paper for setup (c). For other setups, we reproduce FixMatch and OpenMatch using the default configurations. We use 25 and 400 labels for each class. We report the mean and standard deviation for 5 runs for (a) and (b) and 3 runs for (c).

Table 3 presents the results on CIFAR-10. RoPAWS improves PAWS for all scenarios, especially when using extra uncurated data. For example, RoPAWS improves PAWS by +22.7% for the easy OOD (+ Tiny-ImageNet) and +6.9% for the hard OOD (+ CIFAR-100) setups using 25 labels.

---

[9]See Appendix I for additional discussion on why RoPAWS also improves curated data.
[10]See Appendix E for additional discussion on the effect of pre-training methods.

Table 4: Ablation study of each component of RoPAWS, trained on Semi-iNat (uncurated).

|  | Semi-sup density | In-domain prior | Test acc. |
| --- | --- | --- | --- |
| PAWS | - | - | 37.7 |
| RoPAWS (ours) | ✓ | - | 42.5 |
|  | ✓ | ✓ | 43.0 |

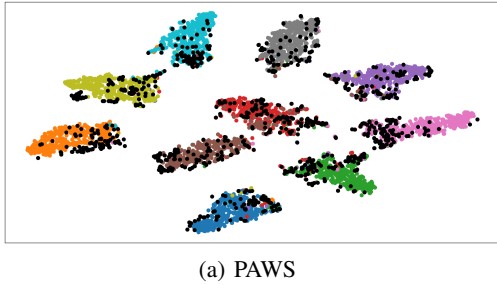 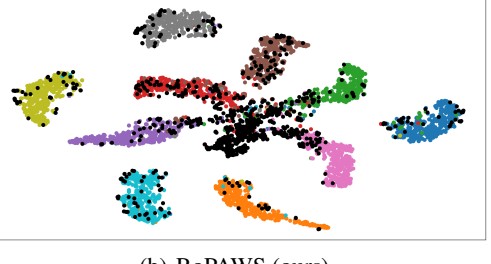

(a) PAWS          (b) RoPAWS (ours)

Figure 4: t-SNE visualization of embeddings of (a) PAWS and (b) RoPAWS trained on CIFAR-10 (+ CIFAR-100) with 400 labels. Colored and black points are from CIFAR-10 and CIFAR-100 respectively. PAWS pushes OOD (CIFAR-100) data to the in-class (CIFAR-10) clusters. In contrast, RoPAWS pushes out-of-class data far from all the in-class clusters.

RoPAWS also exceeds the prior state-of-the-arts: OpenMatch and OpenCos. These OOD filtering methods are effective for easy OOD setups. However, RoPAWS even exceeds OpenCos pre-trained on SimCLR for (+ Tiny-ImageNet) without such pre-training. Also, note that the OOD filtering methods are less effective for hard OOD setups (see Appendix L for additional discussion). RoPAWS is still effective for (+ CIFAR-100) since it does not filter samples but learns from uncurated data. We provide additional comparison with robust Semi-SL methods in Appendix G.

### 4.3 ABLATION STUDY AND ANALYSIS

**Ablation study.** Table 4 presents the ablation study of the components of RoPAWS, trained on Semi-iNat (uncurated). Semi-supervised density estimation in Eq. (11) highly improves the accuracy, and in-domain prior in Eq. (13) (combined with a reweighted loss in Eq. (14)) enhances further. We provide additional ablation studies in Appendix D, showing that RoPAWS gives reasonable results over varying hyperparameters and that all the proposed components are essential.

**Analysis.** We analyze the learned representations of RoPAWS. Figure 4 shows the t-SNE (Van der Maaten & Hinton, 2008) visualization of embeddings, trained on CIFAR-10 (+ CIFAR-100), 400 labels. As PAWS confidently predicts unlabeled data as in-class, OOD data (black points) are concentrated in the in-class clusters (colored points). In contrast, RoPAWS pushes out-of-class data far from all the in-class clusters. We provide additional analyses in Appendix J showing that RoPAWS increases the confidence of in-domain and decreases one of OOD data.

**Nearest neighbors.** We visualize the nearest labeled data of out-of-class data inferred by PAWS and RoPAWS in Appendix F. Even though the classes are different, both models can find visually similar in-class data with slight variation. However, PAWS has high similarity scores to the wrong in-class neighbors, unlike RoPAWS has lower similarities, giving uncertain predictions.

**Computation time.** The inference time of RoPAWS is identical to PAWS. The training time is only slightly increased ($< 5\%$ for each mini-batch) due to the (detached) targets in Eq. (12).

### 5 CONCLUSION

We proposed RoPAWS, a novel robust semi-supervised learning method for uncurated data, which calibrates prediction via density modeling on representation space. We verified the effectiveness of RoPAWS under various scenarios, achieving new state-of-the-art results. In addition, the probabilistic view of RoPAWS can be extended to various problems, as discussed in Appendix M.

ACKNOWLEDGMENTS

This work was supported by Institute of Information & communications Technology Planning & evaluation (IITP) grant funded by the Korea government (MSIT) (No.2019-0-00075, Artificial Intelligence Graduate School Program (KAIST) and No.2021-0-02068, Artificial Intelligence Innovation Hub). SW thanks to Ser-Nam Lim for supporting technical issues for using fairaws cluster.

ETHICS STATEMENT

Learning representations from large and diverse data makes the model more robust and fair (Goyal et al., 2022). However, the uncurated data may inherit the natural bias of data distribution. Thus, the user should not treat the model as a black-box output from arbitrary data but should carefully look at the data's statistics and be aware of the potential biases.

REPRODUCIBILITY STATEMENT

We provide implementation details in Appendix C. We provide our code in the supplementary material, which will be publicly released after the paper acceptance.

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

## A  DETAILED DERIVATIONS

We provide detailed derivations for the prediction of probabilistic PAWS and RoPAWS.

**PAWS (one-hot).** We defined the conditional $p(x|y)$ and marginal $p(x)$ densities as follows:

$$p_{\text{PAWS}}(x|y) := \frac{1}{|\mathcal{B}_l^y|} \sum_{x' \in \mathcal{B}_l^y} k(z, z') \quad \text{and} \quad p_{\text{PAWS}}(x) := \frac{1}{|\mathcal{B}_l|} \sum_{x' \in \mathcal{B}_l} k(z, z'), \tag{15}$$

where $k(z, z') := \frac{1}{Z} \cdot \exp(z \cdot z'/\tau)$. Recall that $|\mathcal{B}_l| = N$, $|\mathcal{B}_l^y| = \frac{N}{C}$, and $p(y) = \frac{1}{C}$.
Note that this definition satisfies the marginalization:

$$p_{\text{PAWS}}(x) = \sum_y p_{\text{PAWS}}(x) \cdot p(y) \tag{16}$$

$$= \sum_y \left( \frac{C}{N} \sum_{x' \in \mathcal{B}_l^y} k(z, z') \right) \cdot \frac{1}{C} \tag{17}$$

$$= \frac{1}{N} \left( \sum_y \sum_{x' \in \mathcal{B}_l^y} \right) k(z, z') \tag{18}$$

$$= \frac{1}{N} \sum_{x' \in \mathcal{B}_l} k(z, z'). \tag{19}$$

Applying the definitions to the Bayes' rule,

$$p_{\text{PAWS}}(y|x) = p_{\text{PAWS}}(x|y) \cdot p(y) \ / \ p_{\text{PAWS}}(x) \tag{20}$$

$$= \frac{C}{N} \sum_{x' \in \mathcal{B}_l^y} k(z, z') \cdot \frac{1}{C} \ / \ \frac{1}{N} \sum_{x^* \in \mathcal{B}_l} k(z, z^*) \tag{21}$$

$$= \sum_{x' \in \mathcal{B}_l^y} \frac{1}{Z} \cdot \exp(z \cdot z'/\tau) \ / \ \sum_{x^* \in \mathcal{B}_l} \frac{1}{Z} \cdot \exp(z \cdot z^*/\tau) \tag{22}$$

$$= \sum_{x' \in \mathcal{B}_l^y} \frac{\exp(z \cdot z'/\tau)}{\sum_{x^* \in \mathcal{B}_l} \exp(z \cdot z^*/\tau)} \tag{23}$$

$$= \sum_{x' \in \mathcal{B}_l^y} \sigma_\tau(z \cdot \mathbf{z}_l^\mathsf{T}) \cdot (\mathbf{1}_{x'}) \tag{24}$$

$$= \sum_{x' \in \mathcal{B}_l^y} \sigma_\tau(z \cdot \mathbf{z}_l^\mathsf{T}) \cdot (\mathbf{1}_{x'} \cdot \mathbb{1}_{x' \in \mathcal{B}_l^y}) \tag{25}$$

$$= \sum_{x' \in \mathcal{B}_l} \sigma_\tau(z \cdot \mathbf{z}_l^\mathsf{T}) \cdot (\mathbf{1}_{x'} \cdot \mathbb{1}_{x' \in \mathcal{B}_l^y}) \tag{26}$$

$$= \sigma_\tau(z \cdot \mathbf{z}_l^\mathsf{T}) \cdot \mathbf{p}_l^{y|x} \tag{27}$$

where $\sigma_\tau(z \cdot \mathbf{z}_l^\mathsf{T}) \in \mathbb{R}^{1 \times N}$ is a similarity vector, $\mathbf{1}_{x'} \in \mathbb{R}^{N \times 1}$ is an index vector where the index of $x'$ is 1 otherwise 0, $\mathbb{1}$ is an indicator function, and $\mathbf{p}_l^{y|x} \in \mathbb{R}^{N \times 1}$ is a (one-hot) label vector.

**PAWS (soft).** Applying the weighted kernel $k^y(z, z') := \frac{1}{K} p(y|x') k(z, z')$, the extended definition of conditional density $p(x|y)$ is as follows:

$$p_{\text{PAWS-soft}}(x|y) := \frac{1}{N} \sum_{x' \in \mathcal{B}_l} k^y(z, z') \tag{28}$$

$$= \frac{1}{NK} \sum_{x' \in \mathcal{B}_l} p(y|x') k(z, z') \tag{29}$$

Then, the marginal density $p(x)$ is given by:

$$p_{\text{PAWS-soft}}(x) := \sum_y p_{\text{PAWS-soft}}(x|y) \cdot p(y) \tag{30}$$

$$= \sum_y \frac{1}{NK} \sum_{x' \in \mathcal{B}_l} p(y|x') k(z, z') \cdot \frac{1}{C} \tag{31}$$

$$= \frac{1}{NK} \sum_{x' \in \mathcal{B}_l} \frac{1}{C} \left( \sum_y p(y|x') \right) k(z, z') \tag{32}$$

$$= \frac{1}{NK} \sum_{x' \in \mathcal{B}_l} \frac{1}{C} k(z, z') \tag{33}$$

Applying the definitions to the Bayes' rule,

$$p_{\text{PAWS-soft}}(y|x) = p_{\text{PAWS-soft}}(x|y) \cdot p(y) \ / \ p_{\text{PAWS-soft}}(x) \tag{34}$$

$$= \frac{1}{NK} \sum_{x' \in \mathcal{B}_l} p(y|x') k(z, z') \cdot \frac{1}{C} \ / \ \frac{1}{NK} \sum_{x^* \in \mathcal{B}_l} \frac{1}{C} k(z, z^*) \tag{35}$$

$$= \sum_{x' \in \mathcal{B}_l} \left( k(z, z') \ / \ \sum_{x^* \in \mathcal{B}_l} k(z, z^*) \right) \cdot p(y|x') \tag{36}$$

$$= \sum_{x' \in \mathcal{B}_l} \left( \exp(z \cdot z'/\tau) \ / \ \sum_{x^* \in \mathcal{B}_l} \exp(z \cdot z^*/\tau) \right) \cdot p(y|x') \tag{37}$$

$$= \sigma_\tau(z \cdot \mathbf{z}_l^\intercal) \cdot \mathbf{p}_l^{y|x} \tag{38}$$

where $\mathbf{p}_l^{y|x} \in \mathbb{R}^{N \times 1}$ is a (soft) label vector.

**RoPAWS.** We extend the densities to consider both labeled and unlabeled data:

$$p_{\text{RoPAWS}}(x|y) = \frac{1}{K'} \sum_{\mathcal{B}_l \cup \mathcal{B}_u} p(y|x')\, k(z, z') \quad \text{and} \quad p_{\text{RoPAWS}}(x) = \frac{1}{K'} \sum_{\mathcal{B}_l \cup \mathcal{B}_u} \frac{1}{C}\, k(z, z'). \tag{39}$$

Here, we balance the effect of labeled and unlabeled data with ratio $r : 1$. Specifically, we use

$$p_{\text{RoPAWS}}(x|y) := \frac{1}{K'} \left( r' \cdot \sum_{x' \in \mathcal{B}_l} p(y|x')\, k(z, z') + \sum_{x' \in \mathcal{B}_u} p(y|x')\, k(z, z') \right) \tag{40}$$

$$p_{\text{RoPAWS}}(x) := \frac{1}{K'} \left( r' \cdot \sum_{x' \in \mathcal{B}_l} \frac{1}{C}\, k(z, z') + \sum_{x' \in \mathcal{B}_u} \frac{1}{C}\, k(z, z') \right) \tag{41}$$

where $r' := r \cdot |\mathcal{B}_u|/|\mathcal{B}_l|$ balances the overall effect of labeled batch. Similar to PAWS soft labels, it satisfies the marginalization and gives a similar prediction formula.

We set $\tilde{r}(x') := r' \cdot \mathbb{1}_{x' \in \mathcal{B}_l} + \mathbb{1}_{x' \in \mathcal{B}_u}$, and simply we can write:

$$p_{\text{RoPAWS}}(x|y) = \frac{1}{K'} \sum_{x' \in \mathcal{B}_l \cup \mathcal{B}_u} \tilde{r}(x') \cdot p(y|x')\, k(z, z') \tag{42}$$

$$p_{\text{RoPAWS}}(x) = \frac{1}{K'} \sum_{x' \in \mathcal{B}_l \cup \mathcal{B}_u} \tilde{r}(x') \cdot \frac{1}{C}\, k(z, z'). \tag{43}$$

Applying the definitions to the Bayes' rule,

$$p_{\text{RoPAWS}}(y|x) = p_{\text{RoPAWS}}(x|y) \cdot p(y)\ /\ p_{\text{RoPAWS}}(x) \tag{44}$$

$$
\begin{aligned}
&= \frac{1}{NK} \sum_{x' \in \mathcal{B}_l} \tilde{r}(x') \cdot p(y|x')\, k(z, z')\ \cdot\ p(y) \\
&\Big/ \frac{1}{NK} \sum_{x^* \in \mathcal{B}_l} \tilde{r}(x^*) \cdot \frac{1}{C}\, k(z, z^*)
\end{aligned}
\tag{45}
$$

$$
\begin{aligned}
&= \sum_{x' \in \mathcal{B}_l \cup \mathcal{B}_u} \left( \tilde{r}(x') \cdot k(z, z')\ /\ \sum_{x^* \in \mathcal{B}_l \cup \mathcal{B}_u} \tilde{r}(x^*) \cdot k(z, z^*) \right) \\
&\cdot p(y|x')\ \cdot\ C\, p(y)
\end{aligned}
\tag{46}
$$

$$
\begin{aligned}
&= \sum_{x' \in \mathcal{B}_l \cup \mathcal{B}_u} \left( \tilde{r}(x') \cdot \exp(z \cdot z'/\tau)\ /\ \sum_{x^* \in \mathcal{B}_l \cup \mathcal{B}_u} \tilde{r}(x^*) \cdot \exp(z \cdot z^*/\tau) \right) \\
&\cdot p(y|x')\ \cdot\ C\, p(y).
\end{aligned}
\tag{47}
$$

Recall that we do not assume $p(y)$ to be uniform nor cancel out with $C$ to control the in-domain prior. This formula can be simplified as a softmax (as above) by adding a constant vector $\mathbf{r}$. Specifically, let $\mathbf{r} \in \mathbb{R}^{1 \times N+M}$ be a vector where the index of $x' \in \mathcal{B}_l$ is $\tau \cdot \log(r')$ and otherwise 0. Then,

$$p_{\text{RoPAWS}}(y|x) = \sigma_\tau(z \cdot (\mathbf{z} + \mathbf{r})^\intercal) \cdot \mathbf{p}^{y|x}\ \cdot\ C\, p(y), \tag{48}$$

and splitting the concatenated vectors $\mathbf{z}$ and $\mathbf{p}^{y|x}$ (and omitting $\mathbf{r}$) into labeled and unlabeled terms gives the final formula:

$$p_{\text{RoPAWS}}(y|x) = \sigma_\tau(z \cdot [\mathbf{z}_l^\intercal \mid \mathbf{z}_u^\intercal]) \cdot [\mathbf{p}_l^{y|x} \mid \mathbf{p}_u^{y|x}]^\intercal\ \cdot\ C\, p(y). \tag{49}$$

To sum up, the prediction of unlabeled data is computed from the predictions of $N$ labeled and $M$ unlabeled data. We stack this for $M$ unlabeled data and then get the matrix formula of prediction for unlabeled data $\mathbf{q} \in \mathbb{R}^{M \times C}$ and labeled data $\mathbf{p} \in \mathbb{R}^{N \times C}$ with weights $\mathbf{a} \in \mathbb{R}^{M \times N}$ and $\mathbf{b} \in \mathbb{R}^{M \times M}$ given by the similarity matrix. Namely, the simplified version of Eq. (10) looks like:

$$\mathbf{q} = \mathbf{a} \cdot \mathbf{p} + \mathbf{b} \cdot \mathbf{q}, \tag{50}$$

and organizing the formula gives the closed-form solution:

$$(\mathbf{I} - \mathbf{b}) \cdot \mathbf{q} = \mathbf{a} \cdot \mathbf{p}, \quad \text{and thus,} \quad \mathbf{q} = (\mathbf{I} - \mathbf{b})^{-1} \cdot \mathbf{a} \cdot \mathbf{p}, \tag{51}$$

which gives the final prediction formula in Eq. (11). Note that $(\mathbf{I} - \mathbf{b})$ is invertible since $\mathbf{b}$ is the part of the similarity matrix, i.e., the row-wise sum of $\mathbf{b} < 1$.

We remark that this formula can be interpreted as the label propagation, i.e., propagating the belief of labels $\mathbf{p}$ (and also the priors on $\mathbf{q}$ in our case) to get the posterior prediction of $\mathbf{q}$. Indeed, think about the iterative formula:

$$\mathbf{q}^{i+1} \leftarrow \mathbf{a} \cdot \mathbf{p} + \mathbf{b} \cdot \mathbf{q}^{i}, \tag{52}$$

for $i \geq 0$ and initialized from the prior $\mathbf{q}^0$. Here, we can think our closed-form solution as the fixed-point of the iteration (it converges at $i \to \infty$ since $\|\mathbf{b}\| < \mathbf{I}$, i.e., contraction mapping).

Finally, recall that the prediction $q(y|x)$ (a row from $\mathbf{q}$) can have a row-wise sum $< 1$ when using the in-domain prior. It implies that the data $x$ is likely to be in-domain with (posterior) probability $\bar{q} = \sum_y q(y|x)$ and OOD with probability $1 - \bar{q}$. We regularize the prediction to be close to uniform for uncertain (i.e., OOD-like) samples. Here, the final prediction value is $q(y|x) + (1 - \bar{q})/C$, which makes the sum of the probability for $C$ in-class to be one.

## B  NON-COLLAPSING PROPERTY OF REPRESENTATION

One of the main advantages of PAWS is non-collapsing representations. RoPAWS also favors this property. Recall that the proof logic of Proposition 1 in the PAWS paper is as follows:

1. When the representation collapses, the distances between representations $d(z, z_i)$ become identical for all $z_i$ in the labeled batch $\mathcal{B}_l$.

2. Then, the prediction $p$ is given by the average of class labels $y_i$, which becomes a uniform distribution due to the class-balanced sampling.

3. However, the target sharpening makes the target $p^+$ not equal to the uniform distribution; hence, $\nabla_\theta H(p^+, p) > 0$, and the model escapes from the representation collapse.

RoPAWS follows the same logic: the prediction $p$ becomes a uniform distribution when representation collapses, being escaped by step 3 above. The difference from PAWS is that now the prediction is the average of true labels $y_i$ of labeled batch $\mathcal{B}_l$ and pseudo-labels $\hat{y}_i$ of unlabeled batch $\mathcal{B}_u$. Here, the initial belief of pseudo-labels $\hat{y}_i$ becomes a uniform distribution by averaging the true labels $y_i$, and averaging this uniform distribution keeps the updated belief uniform after label propagation.

# C  IMPLEMENTATION DETAILS

## C.1  COMMON SETUPS

We follow the default configurations of PAWS. Specifically, we follow the ImageNet configuration for large-scale (Semi-iNat, ImageNet) experiments and the CIFAR-10 configuration for small-scale (CIFAR-10) experiments. We only adopt the number of labels and training epochs for each dataset, as the dataset has a smaller size (longer epochs) or fewer labels (smaller labeled batch).

**Training.** We sample an unlabeled and a labeled batch for each iteration. For the unlabeled batch, we sample 2 views for targets and 6 additional multi-crop views (8 in total) for outputs. We compute the semi-supervised prediction of RoPAWS for each view independently, and the size of the unlabeled batch (so as computation complexity) does not increase over the number of views. We use the softmax temperature $\tau = 0.1$ and sharpen the target with temperature $T = 0.25$, i.e.,

$$\rho(p)_i = \frac{p_i^{1/T}}{\sum_{i=1}^{C} p_i^{1/T}} \tag{53}$$

where $p_i$ and $\rho(p)_i$ denotes the $i$-th element of probability $p, \rho(p) \in \mathbb{R}^{1 \times C}$. We apply the mean entropy maximization (me-max) regularization $-\mathbb{H}(\bar{p})$ for all experiments.

For the labeled batch, we sample $C$ classes (from overall $C_{\text{all}} > C$ classes) for each iteration, where $C$ depends on datasets, e.g., $C = 960$ for ImageNet with $C_{\text{all}} = 1000$. We sample the same number of data for each class (i.e., class balanced) with 1 or 2 augmentation views. Thus, the overall support size is the number of classes × samples per class × number of views.

We apply the same data augmentation with PAWS, SimAug (Chen et al., 2020a), for all experiments. The same augmentation is applied for both unlabeled and labeled data.

For large-scale experiments, we train a ResNet-50 network using an unlabeled batch of size 4096 on 64 GPUs. We use a 3-layer multi-layer perceptron (MLP) projection head with 2048 hidden dimensions and a 2-layer MLP prediction head with 512 hidden dimensions. This projection/prediction head is widely applied for self-supervised learning (Grill et al., 2020). We use the LARS (You et al., 2017) optimizer with momentum value 0.9 and weight decay $10^{-6}$. We linearly warm up the learning rate from 0.3 to 6.4 during the first 10 epochs and apply the cosine schedule afterward.

For small-scale experiments, we train a WRN-28-2 network using an unlabeled batch of size 256 on 1 GPU. We use a 3-layer MLP projection head with 128 hidden dimensions and do not use a prediction head. We use the LARS optimizer with the same configuration of the large-scale setup but linearly warm up the learning rate from 0.8 to 3.2 and then apply the cosine schedule.

**Evaluation.** We use a soft-nearest neighbor classifier with softmax temperature $\tau = 0.1$ (same to the training) for Semi-iNat and CIFAR-10. We fine-tune the classifier upon the learned representation for ImageNet since it gives a further gain. We follow the fine-tuning configuration of PAWS, training a linear classifier on top of the first layer of the projection head initialized from zero weights. We fine-tune the backbone and linear classifier simultaneously, optimizing a supervised cross-entropy loss on the small set of labeled data. We apply the basic data augmentations (random crop and horizontal flip) and sweep the learning rate from $\{0.01, 0.02, 0.05, 0.1, 0.2\}$ and epochs from $\{30, 50\}$, choosing the hyperparameter based on a 12,000 images subset from the ImageNet training set. We use SGD with Nesterov momentum (with value of 0.9) and a batch size of 1024. We apply a single center-crop augmentation for test samples and report the top-1 accuracy.

## C.2  DATASET-SPECIFIC SETUPS

**Semi-iNat.** We train from scratch models for 800 epochs, and ImageNet pre-trained models for 300 epochs. We use 12 classes per batch (thus, $12 \times 64 = 768 < 810$ classes are used for each iteration in total) and 5 images for each class (Semi-iNat is long-tailed, and the tail classes only have 5 labels) with 1 view. We use the official SwAV weights[11] for the ImageNet pre-trained experiments, where we also compare them with the supervised weights[12] used in the prior work. We found

---

[11]`torch.hub.load('facebookresearch/swav:main','resnet50')`
[12]`https://download.pytorch.org/models/resnet50-19c8e357.pth`

that SwAV gives a better initialization for PAWS and RoPAWS as the objective (and thus, learned representations) of SwAV is similar to PAWS, and used SwAV for our main experiments.

**ImageNet.** We train models for 300 epochs, following the default PAWS configuration. We use 15 classes per batch (thus, $15 \times 64 = 960 < 1,000$ classes are used for each iteration in total) and 7 images for each class with 1 view. We use the same subset split of PAWS.

**CIFAR-10.** We train models for 600 epochs, following the default PAWS configuration. We use the full 10 classes and 64 images for each class with 2 views for 400 labels setup. For 25 labels and 4 labels setups, we decrease the images for each class and increase the number of views to keep the total size of labeled support. Specifically, we use 16 images for each class with 8 views for 25 labels setup and 4 images for each class with 32 views for 4 labels setup.

## C.3 Hyperparameters of RoPAWS

RoPAWS has three hyperparameters: the ratio of labeled and unlabeled batch $r$, the temperature for in-domain prior $\tau_{prior}$, and the power of reweighted loss $k$.

**Choice of $r$.** We use $r = 5$ for all experiments following the default batch size ratio in CIFAR-10. CIFAR-10 has a labeled batch size of 10 classes $\times$ 64 samples per class $\times$ 2 views = 1280, and an unlabeled batch size of 256. Thus, the labeled and unlabeled batch ratio is $1280/256 = 5$. We observed that using $r > 1$ is helpful since the prediction on the unlabeled data is uncertain; giving more weights for certain labels is a safer choice. Recall that the original batch ratio of Semi-iNat is $768 \times 5 \times 1/4096 \approx 0.94$, giving more weights for unlabeled than labeled data.

**Choice of $\tau_{prior}$.** We set $\tau_{prior} = 3.0$ for ResNet-50 and $\tau_{prior} = 0.1$ for WRN-28-2. Since ResNet-50 has a larger latent dimension (2048) than WRN-28-2 (128), the embeddings are more likely to differ from the other embeddings, and cosine similarity decreases faster. Thus, we use a larger temperature $\tau_{prior}$ for ResNet-50 to not decay the in-domain prior too rapidly.

**Choice of $k$.** We set $k = 1$ for all experiments except $k = 3$ for fine-tuning experiments on Semi-iNat, using ImageNet pre-trained models. Note that higher $k$ makes the model more conservative; focus on confident samples and ignore uncertain samples. Models from scratch need to learn representation from uncurated data, thus setting lower $k$. In contrast, ImageNet pre-trained models have good representation and correct labels more matter, thus setting higher $k$.

## C.4 Other baselines

We run FixMatch and OpenMatch for CIFAR-10 experiments. Here, we follow the default setup for CIFAR-10 for both models, using the unofficial (but matching the official performance) PyTorch implementation of FixMatch[13] and the official PyTorch implementation of OpenMatch[14]. We did not tune hyperparameters specialized to each setup. However, FixMatch was effective for both curated (CIFAR-10) and hard OOD (CIFAR-10 + CIFAR-100) setups, implying that the default choices are already well-tuned. OpenMatch was effective for easy OOD (400 labels) but did not work well for hard OOD (25 labels) setups on CIFAR-10 (+ CIFAR-100). It can be improved with a better choice of hyperparameters, yet, we remark that the OOD filtering approach (especially the ones discarding samples explicitly) is sensitive to the threshold and is not robust in the OOD setup.

---

[13]https://github.com/LeeDoYup/FixMatch-pytorch
[14]https://github.com/VisionLearningGroup/OP_Match

# D ADDITIONAL ABLATION STUDIES

We provide additional ablation studies for RoPAWS. Table 5 presents the test accuracy of RoPAWS trained on Semi-iNat (uncurated) varying hyperparameters. Specifically, (a) shows the results varying the in-domain $\tau_{\mathrm{prior}}$ while fixing the reweight power $k = 1.0$, and (b) shows the results varying the reweight power $k$ while fixing the in-domain prior $\tau_{\mathrm{prior}} = 3.0$. The results are stable under a reasonable range of hyperparameters, yet tuning may give a further boost.

Table 6 presents an extra ablation study for the components of RoPAWS. We applied reweighted loss when using in-domain prior in Table 4. Indeed, using in-domain prior without reweighted loss harms the accuracy. In addition, only using in-domain prior (and reweighted loss) without semi-supervised density harms the accuracy. It verifies that all the proposed components are essential.

Table 5: Ablation study of hyperparameters of RoPAWS, trained on Semi-iNat (uncurated).

| (a) In-domain prior $\tau_{\mathrm{prior}}$ ($k = 1.0$) | | | (b) Reweight power $k$ ($\tau_{\mathrm{prior}} = 3.0$) | | |
|---|---|---|---|---|---|
| | $\tau_{\mathrm{prior}}$ | Test acc. | | $k$ | Test acc. |
| PAWS | - | 37.7 | PAWS | - | 37.7 |
| RoPAWS (ours) | 1.0 | 42.6 | RoPAWS (ours) | 1.0 | 43.0 |
| | 2.0 | 42.4 | | 2.0 | 42.9 |
| | 3.0 | 43.0 | | 3.0 | 41.1 |
| | 5.0 | 42.5 | | 5.0 | 40.1 |

Table 6: Ablation study of each component of RoPAWS, trained on Semi-iNat (uncurated).

| Semi-sup density | In-domain prior | Reweighted loss | Test acc. |
|---|---|---|---|
| - | - | - | 37.7 |
| - | ✓ | ✓ | 34.4 |
| ✓ | - | - | 42.5 |
| ✓ | ✓ | - | 42.3 |
| ✓ | ✓ | ✓ | 43.0 |

# E EFFECT OF PRE-TRAINING METHOD

Table 7 presents the test accuracy of PAWS and RoPAWS, initialized from ImageNet supervised or SwAV (Caron et al., 2020) pre-trained models. We used SwAV for our main table since it gives a better initialization. Nevertheless, RoPAWS initialized from an ImageNet supervised model is already better than prior arts using the same pre-trained model. Intuitively, the training objective of PAWS (and RoPAWS) more resembles SwAV and thus has less representation shifts.

Table 7: Test accuracy (%) of PAWS and RoPAWS trained on Semi-iNat (uncurated), initialized from ImageNet pre-trained models. SwAV gives a better initialization for PAWS than the supervised model; the loss of PAWS resembles one of SwAV and shifts less the learned representation.

| Pre-trained | Method | Test acc. |
|---|---|---|
| ImageNet supervised | PAWS | 42.0 |
| | RoPAWS (ours) | 42.4 |
| ImageNet SwAV | PAWS | 43.0 |
| | RoPAWS (ours) | 44.3 |

## F VISUALIZATION OF NEAREST NEIGHBORS

Figure 5 visualizes the nearest labeled data and its cosine similarity to queries inferred from PAWS and RoPAWS trained on Semi-iNat (uncurated). Even for the out-of-class data, both models can find visually similar in-class data with slight variation. However, PAWS generates predictions with high confidence (high similarity score) while RoPAWS has lower confidence. For the in-class data, both models generate high confidence predictions and can find the correct labels.

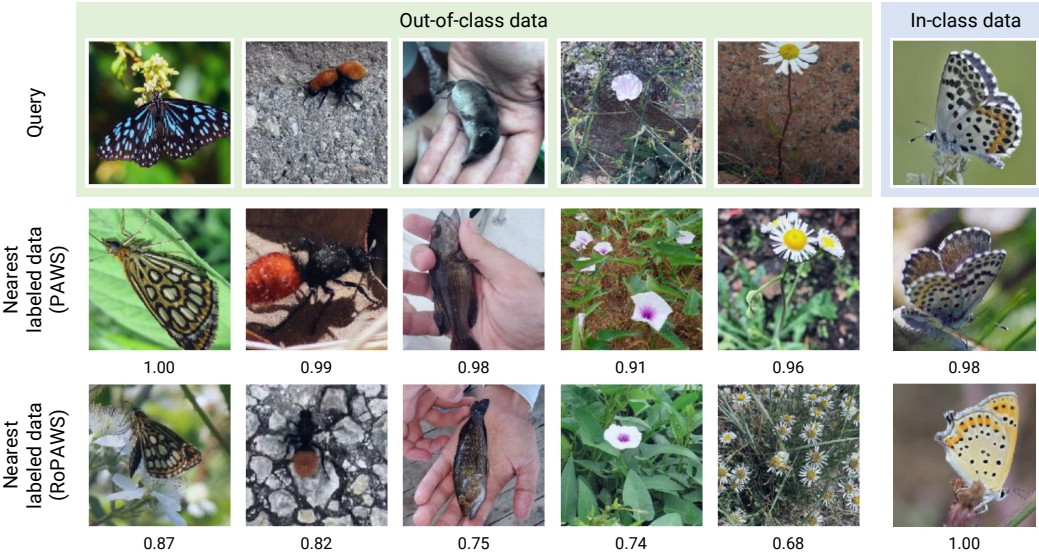

Figure 5: Nearest labeled data and its cosine similarity to queries.

## G ADDITIONAL COMPARISON WITH ROBUST SEMI-SL METHODS

We conduct additional experiments comparing with robust SSL algorithms, using different in-class datasets: (CIFAR-10, CIFAR-100) and out-of-class datasets (SVHN, Tiny-IN). We follow the same configuration of CIFAR-10 experiments in Table 3, except increasing the classes per batch to 100 and unlabeled batch size to 1024 when the in-class dataset is CIFAR-100. We used 25 labels per class, set the labeled batch per class as 16, and reported the average and standard deviation over three runs. The baseline results are from the OpenCos (Park et al., 2021) paper.

Table 8 shows that RoPAWS consistently outperforms the SOTA robust SSL algorithms under different settings. In particular, RoPAWS improves the SOTA accuracy with a large margin (+5.8%) in the most challenging scenario of CIFAR-100 (+ Tiny-IN).

Table 8: Test accuracy (%) of various robust Semi-SL methods under various scenarios. We bold the best results. RoPAWS achieves state-of-the-art results in all considered scenarios.

| In-class | CIFAR-10 | | CIFAR-100 | |
|---|---|---|---|---|
| Out-of-class | SVHN | Tiny-IN | SVHN | Tiny-IN |
| FixMatch (Sohn et al., 2020) | 68.0±0.7 | 70.5±1.2 | 41.7±1.3 | 46.0±1.0 |
| PAWS (Assran et al., 2021) | 47.1±0.4 | 60.4±5.3 | 36.9±0.2 | 46.0±0.5 |
| UASD (Chen et al., 2020d) | 66.7±1.0 | 74.0±0.4 | 39.5±0.8 | 44.6±0.8 |
| RealMix (Nair et al., 2019) | 58.2±5.3 | 69.2±2.3 | 44.1±1.0 | 47.6±1.4 |
| DS$^3$L (Guo et al., 2020) | 50.0±8.3 | 69.1±2.3 | 29.0±1.0 | 40.2±0.9 |
| OpenMatch (Saito et al., 2021) | 74.7±1.0 | 78.0±0.8 | 29.0±1.3 | 32.2±2.6 |
| OpenCoS (Park et al., 2021) | 79.0±1.1 | 82.5±1.3 | 49.2±1.2 | 54.0±1.8 |
| RoPAWS (ours) | **81.2**±0.4 | **83.1**±1.8 | **51.5**±1.7 | **59.8**±1.3 |

# H    COMPARISON WITH SSKDE

SSKDE (Wang et al., 2009) also compute the posterior probability of labels based on the KDE over the feature space. However, SSKDE has several critical limitations:

- SSKDE requires an ad-hoc feature extractor. SSKDE applies KDE to the precomputed features, unlike RoPAWS jointly learning the representation and the KDE generative classifier. The ability to learn good representations is one of the main differences between RoPAWS and pre-deep-learning approaches like SSKDE.

- SSKDE is infeasible for large-scale data. Since SSKDE is a nonparametric method, it computes the kernel matrix between all training data. Thus, the training cost of SSKDE is $O(N^3)$, where $N$ is the number of training data. In contrast, RoPAWS only computes the kernel matrix within the mini-batches, reducing the training cost to $O(N * M^3)$, where M is the size of the mini-batch and $M \ll N$. This enables the training under arbitrarily large data.

Due to the limitations listed above, SSKDE cannot be applied to real-world large-scale datasets without significant modification. In addition, RoPAWS has multiple technical novelties over SSKDE:

- The target problem is different. The main goal of RoPAWS is robust semi-supervised learning under uncurated data. Thus, we propose in-domain prior to handling out-of-distribution (OOD) samples. In contrast, SSKDE aims for the curated setup and does not address the overconfidence issue of OOD samples. In addition, SSKDE mainly targets the video annotation problem and only conducts small-scale experiments, while RoPAWS verifies its effectiveness on realistic large-scale image benchmarks.

- The definition of kernel is different. RoPAWS discovers that PAWS is implicitly modeling KDE with cosine similarity kernels, where the kernel definition comes from the design of recent self-supervised learning methods. In contrast, SSKDE uses Gaussian or Exponential kernels, which may not fit with deep learning techniques.

- The formulation and required assumption are different. RoPAWS leverages the class-balanced sampling of PAWS, canceling out the denominator of $p(x|y)$ in the Bayes rule. SSKDE considers a different assumption that $p(y|x)$ is a linear combination of $\hat{p}(y|x)$ and ground-truth labels (if exists), where $\hat{p}(y|x)$ is again a function of $p(y|x)$ and kernels. As a result, the final prediction formula of SSKDE in Eq. (14) of their paper and RoPAWS in Eq. (11) are different.

Despite the fundamental limitations of SSKDE, we provide an experiment comparing RoPAWS and SSKDE. Since SSKDE requires an ad-hoc feature extractor, we run SSKDE upon PAWS and RoPAWS backbones, in addition to a randomly initialized linear layer of dimension 3x32x32→128. We follow the default configuration of SSKDE: use the Gaussian kernel with gamma = 1/{# of features}, weight for the linear interpolation $t = 0.9$, and number of iterations $M = 40$.

Table 9 shows the test accuracy models on CIFAR-10 using 25 labels per class. First, one can see that the quality of the feature extractor mostly decides the final accuracy. We remark that RoPAWS learns a good representation, which is a clear advantage over SSKDE. Second, the Soft-NN classifier (default of PAWS/RoPAWS) performs better than the SSKDE classifier. Indeed, the representation of PAWS/RoPAWS is optimized for the Soft-NN classifier. The results confirm that RoPAWS is empirically better than SSKDE. However, exploring different designs of generative classifiers (e.g., extending SSKDE to learn representations jointly) would be an interesting future research direction.

Table 9:   Test accuracy of RoPAWS and SSKDE under different feature extractors. The quality of representation mostly decides the final accuracy. Upon the PAWS/RoPAWS representation, the Soft-NN classifier performs better than SSKDE, as PAWS/RoPAWS are optimized for this. Note that SSKDE only provides the inference method and does not learn the corresponding feature extractor.

| Feature extractor | Random linear | PAWS | | RoPAWS | |
|---|---|---|---|---|---|
| Inference | SSKDE | SSKDE | Soft-NN | SSKDE | Soft-NN |
| Test acc. | 13.9 | 91.0 | 92.4 | 92.8 | 94.7 |

## I   IMPROVEMENTS ON CURATED DATA

RoPAWS improves PAWS for curated data since it refines the pseudo-labels via label propagation. This benefit is more significant when the original pseudo-label of PAWS is inaccurate, e.g., only a few labels are available. As a result, RoPAWS gives a large gain (+2.3%) for CIFAR-10 using 25 labels. However, the gap saturates when using more labels, e.g., 250 labels for CIFAR-10 or 10K (1%) or 100K (10%) labels for ImageNet. Table 10 shows the test accuracy of models trained on CIFAR-10 using 4 labels. We report the average and standard deviation over five runs. RoPAWS highly improves (+3.0%) PAWS in this more challenging scenario.

Table 10:  Test accuracy (%) of semi-supervised learning methods trained on curated CIFAR-10.

|          | FixMatch | PAWS | RoPAWS (ours) |
|----------|----------|------|---------------|
| 4 labels | $86.2_{\pm 3.4}$ | $85.7_{\pm 2.4}$ | $88.7_{\pm 2.2}$ (+2.3) |

We provide additional analyses verifying the benefit of RoPAWS. Specifically, RoPAWS *reduces confirmation bias* (Arazo et al., 2020) and *improves calibration* (Loh et al., 2022) of pseudo-labels. Recall that lowering wrong predictions (i.e., less confirmation bias) and understanding right confidence (i.e., better calibration) play a crucial role in label propagation.

First, we verify that label propagation refines the pseudo-labels and reduces confirmation bias (i.e., improves correctness). To this end, we present the accuracy of pseudo-labels over label propagation, following the same setup of Appendix K. Table 11 shows the test accuracy of RoPAWS trained on curated CIFAR-10 using 25 labels, using different epochs of checkpoints (total: 600) and different soft nearest-neighbor (Soft-NN) classifiers. Iter. 0 means the Soft-NN classifier only uses labeled data, and Iter. $k \geq 1$ means the iterations of label propagation (using both labeled and unlabeled data) where $k = \infty$ is the fixed-point solution. The pseudo-label becomes more accurate as iteration goes on. Also, note that the effect of label refinement is more impactful in early epochs, where the pseudo-labels are inaccurate. This correction makes RoPAWS converge to better optima.

Table 11:  Test accuracy (%) of a RoPAWS model trained on curated CIFAR-10 using 25 labels. We compare the different inference strategies: the original soft-NN and the label propagation version varying the number of iterations. Label propagation improves the accuracy. Also, note that the effect of label refinement is more impactful in early epochs when the pseudo-labels are inaccurate.

| Epoch | Iter. 0 | Iter. 1 | Iter. 2 | Iter. 3 | Iter. $\infty$ |
|-------|---------|---------|---------|---------|----------------|
| 10    | 73.8    | 74.6    | 74.6    | 75.0    | 75.0           |
| 100   | 87.9    | 88.7    | 88.7    | 88.7    | 88.7           |
| 600   | 95.0    | 95.3    | 95.3    | 95.3    | 95.3           |

Second, we compare the calibration of PAWS and RoPAWS classifiers. To this end, we measure expected calibration error (ECE) (Loh et al., 2022). We apply the TorchMetric implementation[15] and use the default 15 bins. Table 12 shows the ECE (lower is better) of models trained on CIFAR-10 using 25 labels. RoPAWS gives better calibration than PAWS.

Table 12:  ECE (%, lower is better) of models trained on curated CIFAR-10 using 25 labels.

| PAWS | RoPAWS (ours) |
|------|---------------|
| 1.52 | 0.92          |

---

[15]https://torchmetrics.readthedocs.io/en/v0.10.2/classification/calibration_error.html

## J ADDITIONAL ANALYSES FOR ROPAWS

Table 13 shows the prediction confidence and AUROC for PAWS and RoPAWS, trained on CIFAR-10 (+ CIFAR-100) using 25 or 400 labels. AUROC is a popular metric for OOD detection, measuring the area under the receiver operating characteristic (ROC) curve (Bradley, 1997). Precisely, the ROC curve gives the trade-off between true/false positive/negative varying the threshold for OOD detection, and AUROC computes the overall OOD discriminability over thresholds.

We use the soft nearest-neighbor classifier for prediction $p(y|x)$ and maximum representation similarity to labeled data $s(x) := \max_{x_l \in \mathcal{B}_l} z \cdot z_l$ for OOD detection (low $s(x)$ = OOD). PAWS predicts CIFAR-10 (in) and CIFAR-100 (out) with a similar confidence level. In contrast, RoPAWS predicts CIFAR-10 with high confidence while assigning low confidence for OOD data.

Table 13: Prediction confidence (%) and AUROC (%) of PAWS and RoPAWS trained on CIFAR-10 (+ CIFAR-100). RoPAWS predicts CIFAR-10 (in) with high confidence while calibrating prediction for CIFAR-100 (out). In contrast, PAWS assigns mid-level confidence for both data.

|  | 25 labels | | | 400 labels | | |
|---|---|---|---|---|---|---|
|  | Conf. (in) | Conf. (out) | AUROC | Conf. (in) | Conf. (out) | AUROC |
| PAWS | 87.1 | 85.2 | 70.8 | 94.0 | 90.9 | 81.2 |
| RoPAWS (ours) | 92.4 | 65.6 | 81.5 | 95.0 | 67.1 | 87.0 |

## K EFFECT OF LABEL PROPAGATION

We compare the original soft-NN classifier using only labeled data (Eq. (1)) and the semi-supervised (label propagation) version using both labeled and unlabeled data (Eq. (52); becomes Eq. (12) at Iter. $\infty$). Table 14 shows the effect of label propagation during inference, using the RoPAWS model trained on CIFAR-10 (+ CIFAR-100), 400 labels. We sample a batch of labeled and unlabeled data and evaluate the accuracy of unlabeled (in) data, along with the prediction confidence of unlabeled (in) and unlabeled (out) data. The semi-supervised soft-NN calibrates the prediction and improves the accuracy as epochs go, outperforming the original soft-NN. It converges after 3 iterations, so one may apply the iterative formula instead of the matrix inversion for RoPAWS prediction.

Table 14: Test accuracy (%) and prediction confidence (%) on CIFAR-10 (in) and CIFAR-100 (out), using a RoPAWS model trained on CIFAR-10 (+ CIFAR-100), 400 labels. We compare the different inference strategies: the original soft-NN, and the label propagation version varying the number of iterations. Label propagation calibrates the prediction and improves the accuracy.

|  | Iter. | Test acc. | Conf. (in) | Conf. (out) |
|---|---|---|---|---|
| Soft-NN | - | 93.8 | 93.9 | 67.3 |
| Soft-NN (semi-sup) | 1 | 93.8 | 76.8 | 42.6 |
|  | 2 | 94.1 | 84.2 | 46.7 |
|  | 3 | 94.5 | 85.0 | 47.2 |
|  | $\infty$ | 94.5 | 85.1 | 47.2 |

## L   OPENMATCH UNDER SMALL DISTRIBUTION SHIFTS

OOD filtering approaches are less effective when in- vs. out-of-domain samples are not discriminative. Since they cannot identify which data is OOD, they conservatively select training samples hence the model is optimal. Consequently, OpenMatch performs well for CIFAR-10 (+ CIFAR-100) using 400 labels but underperforms when using 25 labels. Table 15 shows that OpenMatch fails to detect OOD samples for 25 labels setup, leading to a meager selected ratio. Also, the OOD filtering approaches underperform on curated datasets since it filters uncertain in-class samples.

Table 15:   Selected ratio (%) of confident samples and AUROC (%) of CIFAR-10 vs. CIFAR-100 for the OpenMatch models trained on CIFAR-10 and CIFAR-10 (+ CIFAR-100). The model cannot distinguish unseen in-class (not covered by a few labels) and near out-of-class samples, as shown by AUROC. As a result, the model filters too many samples and is under-trained.

|  | CIFAR-10 | | CIFAR-10 (+ CIFAR-100) | |
| --- | --- | --- | --- | --- |
|  | 25 labels | 400 labels | 25 labels | 400 labels |
| Selected ratio | 27.5 | 98.9 | 0.2 | 37.4 |
| AUROC | - | - | 54.2 | 91.5 |

## M   FUTURE RESEARCH DIRECTIONS

We believe the probabilistic formulation of RoPAWS, i.e., KDE modeling view over representation space, has the potential to be extended to various scenarios in a principled manner.

For example, we assume the prior class distribution $p(y)$ is uniform over all classes. One can extend RoPAWS to class-imbalanced setup (Kim et al., 2020; Wei et al., 2021) by assuming non-uniform prior. RoPAWS can also handle label shift, i.e., class distribution of labeled and unlabeled data are different. Indeed, the prediction of RoPAWS only assumes the prior on unlabeled data, which can be controlled independently from the prior on labeled data. On the other hand, the entropy mean entropy maximization (me-max) regularization $-\mathbb{H}(\bar{p})$ in the objective is identical to minimizing the KL divergence between the mean prediction and the uniform prior. This can be extended to minimizing the (posterior) mean prediction and the prior when using non-uniform prior.

Also, one can extend RoPAWS to open-world setup (Cai et al., 2022; Vaze et al., 2022), which clusters novel classes in addition to predicting known classes. Here, one can model conditional densities $p(x|y)$ for novel classes $y \in \{C+1, ..., C+C_{\text{novel}}\}$. Then, finding the maximum likelihood assignment of cluster $y$ for unlabeled data would cluster both labeled and unlabeled data similarly to the infinite Gaussian mixture model (Rasmussen, 1999) but under KDE cluster modeling.

Finally, one may use the estimated probability, e.g., $p(\text{in}|x)$ or $p(x|y)$, to choose the most uncertain sample to label for active learning (Settles, 2009). Combining semi-supervised learning and active learning under large uncurated data would be important for practical scenarios.

We remark that the principle of RoPAWS can be generalized beyond PAWS and image classification. For example, the standard softmax classifier can be interpreted as a generative classifier modeling densities with the Gaussian mixture model (GMM) (Lee et al., 2018b). Thus, one can adapt the softmax classifier of previous algorithms, such as FixMatch (Sohn et al., 2020), as a GMM classifier to estimate the uncertainty of samples. However, GMM needs sufficient labels to estimate the variance of each class; hence, less fits for semi-supervised learning.

One may extend RoPAWS for non-classification tasks such as object detection (Liu et al., 2022) by applying dense self-supervision (Wang et al., 2021) or object-aware augmentation (Mo et al., 2021). Also, one may tackle non-image domains, using different augmentations (Lee et al., 2021).

