# OpenReview forum: "RoPAWS: Robust Semi-supervised Representation Learning from Uncurated Data"
_ICLR.cc/2023/Conference — ICLR 2023 poster_

### Official Review · Reviewer_oxEy · 2022-10-23

**Confidence:** 3
**Correctness:** 4
**Technical Novelty And Significance:** 2
**Empirical Novelty And Significance:** 2
**Recommendation:** 8

**Clarity, Quality, Novelty And Reproducibility:**

The paper is well written and easy to follow. It appears to be technically sound and gives adequate theoretical explanation and experiments.


**Strength And Weaknesses:**

Strengths

- Probabilistic explanation of PAWS that it works as generative classifier and extension to RoPAWS make sense and interesting.
- The problem definition and formulation were clear.
- Straightforward solution for the given limitation (overconfident prediction).
- It obtains SOTA results with moderate additional time cost.
- Fig. 4 shows apparently how RoPAWS works effectively compared to PAWS

Weaknesses

- The performances of the proposed RoPAWS as well as PAWS will be highly dependent on the choice of data augmentation methods for creating different views. The details are missing.
- This is rather a question: I wonder if the self-supervisory signals between the two views of the labeled data gives better performance.



**Summary Of The Paper:**

This paper argues that the PAWS is not robust to unlabeled real-world data. This paper tries to tackle this problem by suggesting the RoPAWS. RoPAWS uses in-domain prior to prevent a model from overconfidently predicting unseen class data by adopting regularization utilizing similarity between labeled and unlabeled data. Thereby RoPAWS shows better performance in various settings. RoPAWS especially excels in uncurated settings with fewer labels.
Also, this paper gives a novel probabilistic view of PAWS.



**Summary Of The Review:**

The motivation of RoPAWS is well supported. I believe that sufficient advances in the performance on uncurated settings shows the effectiveness of RoPAWS. Also the analysis using t-SNE demonstrates that RoPAWS works as the authors intended.

---

> ### Author Response · Authors · 2022-11-11
> **Response to Reviewer oxEy**
>
> Dear Reviewer oxEy,
>
> Thank you for your valuable feedback and comments.
>
> ---
>
> **Choice of data augmentation**
>
> We follow the same data augmentation scheme with PAWS: apply SimAug for both labeled and unlabeled data. We clarified this in Appendix A of our revised manuscript. Nevertheless, we provide an additional experiment comparing different augmentations for the readers’ interest. The table below shows the test accuracy of models trained on CIFAR-10 (+ CIFAR-100) using 25 labels per class, using weak (only RandomCrop+HorizontalFlip) or strong (RandAug) augmentations from FixMatch. PAWS/RoPAWS needs sufficiently strong augmentation, such as color distortion, which is the only difference between SimAug over FixMatch (weak).
>
> \begin{array}{lccc}
> & \text{SimAug} & \text{FixMatch (weak)} & \text{FixMatch (strong)} \newline
> \hline
> \text{PAWS} & 78.7 & 61.8 & 78.1 \newline
> \text{RoPAWS (ours)} & 85.6 & 76.2 & 84.8 \newline
> \end{array}
>
> ---
>
> **Will self-supervisory signals between the two views of the labeled data give better performance?**
>
> PAWS already gives the self-supervisory signals for labeled data since it defines the labeled data as $\mathcal{D}_l$ as a subset of the entire data $\mathcal{D}_u$. For example, when using 1% of labels for ImageNet, the unlabeled data $\mathcal{D}_u$ is the full 100% of ImageNet instead of the remaining 99%. We clarified this in the problem setup section of our revised manuscript.
>
> However, we note that the effect of labeled data is negligible as it only takes a small portion of unlabeled data (e.g., 25 labels per class for CIFAR-10 = 25/5000 = 0.5%). We provide an additional experiment verifying the effect of labeled data. The table below shows the test accuracy of models trained on CIFAR-10 (+ CIFAR-100) using 25 labels per class, with labeled data ($\mathcal{D}_u$) and without labeled data ($\mathcal{D}_u \setminus \mathcal{D}_l$). Both cases give similar accuracy under standard deviation.
>
> \begin{array}{lcc}
> & \text{With labeled data ($\mathcal{D}_u$)} & \text{Without labeled data ($\mathcal{D}_u \setminus \mathcal{D}_l$)} \newline
> \hline
> \text{PAWS} & 78.7\pm{2.9} & 76.6 \newline
> \text{RoPAWS (ours)} & 85.6\pm{3.0} & 85.7 \newline
> \end{array}
>
> ---
>
> Sincerely,\
> Authors

---

### Official Review · Reviewer_NjZV · 2022-10-24

**Confidence:** 5
**Clarity, Quality, Novelty And Reproducibility:** 1) The paper is clear and easy to und…
**Correctness:** 2
**Technical Novelty And Significance:** 2
**Empirical Novelty And Significance:** 2
**Recommendation:** 5

**Strength And Weaknesses:**

Strength:
1) This paper focuses on the robust semi-supervised learning problem with out-of-class unlabeled data. This is an important problem.
2) The proposal achieves better performance compared with PAWS.

Weakness:
1) This paper only gives a minor improvement on the PAWS algorithm. The contribution and novelty are quite limited. Can the proposal be applied to the general semi-supervised learning algorithms?
2) There are also some other robust SSL algorithms that consider the out-of-class unlabeled data. More related methods should be compared in the experiments.


---------Update-------
I notice that the authors compared with some robust SSL algorithms on CIFAR-10 datasets. I recommend the authors conduct experiments on more settings, such as different datasets, various labels, various extents of OOD unlabeled data, etc.


**Summary Of The Paper:**

This paper aims to make semi-supervised learning robust when unlabeled data contains out-of-class data. This is a practical problem. The authors improve the semi-supervised learning algorithm PAWS by calibrating the prediction of PAWS based on the data densities. Experimental results show the proposal achieves better performance compared with the PAWS algorithm.

**Summary Of The Review:**

Overall, the contribution and novelty of this paper is limited, the experimental results are also not convincing.

---

> ### Author Response · Authors · 2022-11-11
> **Response to Reviewer NjZV**
>
> Dear Reviewer NjZV,
>
> Thank you for your valuable feedback and comments.
>
> ---
>
> **Minor improvement over PAWS**
>
> RoPAWS significantly improves PAWS, increasing the accuracy by +5.3% on Semi-iNat, +6.9% on CIFAR-10 (+ CIFAR-100) and +22.7% on CIFAR-10 (+ Tiny-IN) using 25 labels per class. As other reviewers highlighted, RoPAWS gives a simple fix for PAWS making it more applicable to real-world problems. In addition to these uncurated scenarios, RoPAWS consistently improves PAWS in curated setups, especially given only a few labels (hence, the pseudo-labels are often inaccurate, and one should consider their uncertainty). For example, Table 3a in our paper shows that RoPAWS improves PAWS by +2.3% for CIFAR-10 using 25 labels per class. In addition, we provide a new experiment on CIFAR-10 using 4 labels per class. The table below shows the test accuracy of models, reporting the average and standard deviation over five runs. RoPAWS gives a large improvement (+3.0%) over PAWS.
>
> \begin{array}{ccc}
> \text{FixMatch} & \text{PAWS} & \text{RoPAWS (ours)} \newline
> \hline
> 86.2\pm{3.4} & 85.7\pm{2.4} & 88.7\pm{2.2}
> \end{array}
>
> In addition, our principle of using a generative classifier for robust semi-supervised learning suggests a general direction for future methods. The concept of semi-supervised density estimation (to calibrate the prediction) in Eq. (11) and in-domain prior (to consider OOD samples) in Eq. (12) can be applied to arbitrary algorithms by adapting the definition of density. For example, the standard softmax classifier can be interpreted as a generative classifier modeling densities with the Gaussian mixture model (GMM) [1]. Thus, one can adapt the softmax classifier of previous algorithms, such as FixMatch, as a GMM classifier to estimate the uncertainty of samples. However, GMM needs sufficient labels to estimate the variance for each class; hence, less fits for semi-supervised learning, as we discussed on page 5 of our paper. We added this discussion in Appendix B of our revised manuscript.
>
> [1] Lee et al. A Simple Unified Framework for Detecting Out-of-Distribution Samples and Adversarial Attacks. NeurIPS 2018.
>
> ---
>
> **Comparison with other Robust SSL methods**
>
> We compare RoPAWs with various robust SSL methods, including CCSSL [1] for Semi-iNat (Table 1), and UASD [2], DS$^3$L [3], OpenMatch [4], and OpenCos [5] for CIFAR-10 (Table 3). To our best knowledge, CCSSL (CVPR 2022) is the SOTA for Semi-iNat, and OpenMatch (NeurIPS 2021) and OpenCos (ECCVW 2022) are the SOTA for CIFAR-10. Please let us know if there are any specific missing related works.
>
> [1] Yang et al. Class-Aware Contrastive Semi-Supervised Learning. CVPR 2022.\
> [2] Chen et al. Semi-Supervised Learning under Class Distribution Mismatch. AAAI 2020.\
> [3] Guo et al. Safe Deep Semi-Supervised Learning for Unseen-Class Unlabeled Data. ICML 2020.\
> [4] Saito et al. OpenMatch: Open-set Consistency Regularization for Semi-supervised Learning with Outliers. NeurIPS 2021.\
> [5] Park et al. OpenCoS: Contrastive Semi-supervised Learning for Handling Open-set Unlabeled Data. ECCV Workshop on Learning from Limited and Imperfect Data, 2022.
>
> ---
>
> Sincerely,\
> Authors

---

> > ### Comment · Reviewer_NjZV · 2022-11-17
> > **Thanks for the response**
> >
> > Thanks for the authors' response. I will raise the score.

---

> > > ### Author Response · Authors · 2022-11-18
> > > **Response to the Updated Review**
> > >
> > > Dear Reviewer NjZV,
> > >
> > > Thank you for your suggestion. We would like to kindly remind you that we provided results on Semi-iNat [1] (not only CIFAR-10) in Table 1, showing that RoPAWS outperforms the SOTA Robust Semi-SL method, CCSSL [2]. Nevertheless, as you suggested, we will conduct more experiments compared with prior robust Semi-SL methods. Since the discussion period is ending soon, we will update the results in the comments and include them in the final manuscript.
> > >
> > > [1] Su & Maji. The Semi-Supervised iNaturalist Challenge at the FGVC8 Workshop. arXiv 2022.\
> > > [2] Yang et al. Class-Aware Contrastive Semi-Supervised Learning. CVPR 2022.
> > >
> > > Sincerely,\
> > > Authors

---

> > > ### Author Response · Authors · 2022-11-27
> > > **Additional comparison with Robust SSL algorithms**
> > >
> > > Dear Reviewer NjZV,
> > >
> > > We conduct additional experiments comparing with robust SSL algorithms, using different in-class datasets (CIFAR-10, CIFAR-100) and out-of-class datasets (SVHN, Tiny-IN). We follow the same configuration of CIFAR-10 experiments in Table 3, except increasing the classes per batch to 100 and unlabeled batch size to 1024 when the in-class dataset is CIFAR-100. We used 25 labels per class, set the labeled batch per class as 16, and reported the average and standard deviation over three runs. The baseline results are from the OpenCos [1] paper. The table below shows that RoPAWS consistently outperforms the SOTA robust SSL algorithms under different settings. In particular, RoPAWS improves the SOTA accuracy with a large margin (+5.8\%) in the most challenging scenario of CIFAR-100 (+ Tiny-IN). We will include the table in the final manuscript.
> > >
> > > \begin{array}{l|cc|cc}
> > > \text{In-class} & \text{CIFAR-10} & \text{CIFAR-10} & \text{CIFAR-100} & \text{CIFAR-100} \newline
> > > \text{Out-of-class} & \text{SVHN} & \text{Tiny-IN} & \text{SVHN} & \text{Tiny-IN} \newline
> > > \hline
> > > \text{FixMatch}      & 68.0\pm0.7 & 70.5\pm1.2 & 41.7\pm1.3 & 46.0\pm1.0 \newline
> > > \text{PAWS}          & 47.1\pm0.4 & 60.4\pm5.3 & 36.9\pm0.2 & 46.0\pm0.5 \newline
> > > \hline
> > > \text{UASD}          & 66.7\pm1.0 & 74.0\pm0.4 & 39.5\pm0.8 & 44.6\pm0.8 \newline
> > > \text{RealMix}       & 58.2\pm5.3 & 69.2\pm2.3 & 44.1\pm1.0 & 47.6\pm1.4 \newline
> > > \text{DS$^3$L}       & 50.0\pm8.3 & 69.1\pm2.3 & 29.0\pm1.0 & 40.2\pm0.9 \newline
> > > \text{OpenMatch}     & 74.7\pm1.0 & 78.0\pm0.8 & 29.0\pm1.3 & 32.2\pm2.6 \newline
> > > \text{OpenCoS}       & 79.0\pm1.1 & 82.5\pm1.3 & 49.2\pm1.2 & 54.0\pm1.8 \newline
> > > \text{RoPAWS (ours)} & 81.2\pm0.4 & 83.1\pm1.8 & 51.5\pm1.7 & 59.8\pm1.3 \newline
> > > \end{array}
> > >
> > > [1] Park et al. OpenCoS: Contrastive Semi-supervised Learning for Handling Open-set Unlabeled Data. ECCV Workshop on Learning from Limited and Imperfect Data, 2022.
> > >
> > > Sincerely,\
> > > Authors

---

> > > > ### Author Response · Authors · 2022-12-05
> > > > **Multi-seed results**
> > > >
> > > > Dear Reviewer NjZV,
> > > >
> > > > We updated the table with three seed results following the prior work. RoPAWS significantly improves PAWS (+14$\sim$34\%) and consistently outperforms the prior SOTA robust SSL methods. For example, RoPAWS improves OpenCos by +2.2\% for CIFAR-10 (+ SVHN) and CIFAR-100 (+ SVHN) and by +5.5\% for the most challenging scenario of CIFAR-100 (+ Tiny-IN).
> > > >
> > > > Sincerely,\
> > > > Authors

---

### Official Review · Reviewer_Q9GX · 2022-10-24

**Confidence:** 3
**Clarity, Quality, Novelty And Reproducibility:** + The paper is well written, good mot…
**Correctness:** 3
**Technical Novelty And Significance:** 3
**Empirical Novelty And Significance:** 2
**Recommendation:** 8

**Strength And Weaknesses:**

+ Quite simple extensions to pave the way for real world problems.
+ Experiments a thoughtful and support the main aim of the approach.


**Summary Of The Paper:**

The paper at hand prososes an extension of a recent proposed semi-supervised representation learning approach PAWS. The extension focuses in robustness, hence, RoPAWS, concerning uncurated data from, e.g., out-of-distribution data. This is a relevant problem in many real world applications.

**Summary Of The Review:**

The paper addresses a relevant problem and proposes an extension to "fix" limitations of the recently proposed PAWS method. The extensions seems incremental but I'm still in favor for the paper and think its worth publishing.

---

> ### Author Response · Authors · 2022-11-11
> **Response to Reviewer Q9GX**
>
> Dear Reviewer Q9GX,
>
> We appreciate your positive comments and a nice summary of our manuscript.
>
> As you highlighted, our proposed RoPAWS provides a simple “fix” of the previous SOTA method PAWS being more robust on real-world uncurated data, supported by thoughtful experiments.
>
> We believe our work would extend the applicability of self-supervised learning to real-world problems.
>
> Sincerely,\
> Authors

---

### Official Review · Reviewer_R1Ae · 2022-10-25

**Confidence:** 3
**Correctness:** 3
**Technical Novelty And Significance:** 3
**Empirical Novelty And Significance:** 3
**Recommendation:** 8

**Clarity, Quality, Novelty And Reproducibility:**

The text is well written and related works are well contextualized. The experiments are fairly complete and clearly explained.

The method does provide novelty in the re-framing and modification of an existing method to allow for better utilizing uncurated unlabeled data.

The text provides enough detail for reproducibility and authors provide their code.

**Strength And Weaknesses:**

Strengths:
*  The work reframes the PAWS method as a generative classifier and uses this probabilistic interpretation to clearly introduce the calibration using the unlabeled data in this setting.
* The method prevents over-confident class predictions for OOD data in the unlabeled set, unlike PAWS, which yields better representations for classification.
* Ablation study is provided to provide a careful study of the components of the method
* Sensitivity of hyper parameters are empirically investigated and show improvements over baseline PAWS for a range of hyperparameter settings.


Weaknesses:
* In the original PAWS, class-balanced sampling plays an important role to avoid collapse of representations. By introducing the unlabeled data, it is not clear if this trivial collapse is now a problem in RoPAWS. Can this point be clarified in the exposition of the method?

* When applying this method to curated data, it is not clear any improvements in performance should be expected. The model will be learning from more data, but the labels inferred for unlabeled data may not be accurate and therefore hurt performance. Indeed the experiments show only minor, perhaps negligible, improvement over PAWS.  The text does claim superiority in these cases as well but the numbers do not generally support the claim. The performance seems generally equal. An interesting point to investigate is the exception to this pattern where RoPAWS conclusively improves on PAWS in the curated CIFAR-10 dataset with only 25 labels per class. Here RoPAWS does benefit from using the (in-distribution) unlabeled data. This could be a strength of the proposed method if investigated further.

* Likewise, the results on ImageNet (Table 2) show similarly small improvements in performance of RoPAWS over PAWS at both 1% and 10% of labels.  Some discussion of this could improve the empirical analysis.

**Summary Of The Paper:**

This paper extends the semi-supervised image classifier PAWS (Assran et al., 2021) by using unlabeled data in addition to the labeled data to inform the representation. Using the unlabeled data in this way allows the model to learn from out-of-distribution classes rather than assuming unlabeled data is curated to contain the same classes seen in labeled data. Experimental results show that the proposed RoPAWS method performs roughly on-par with baseline PAWS method on curated datasets and improves on PAWS in uncurated (OOD) settings.

**Summary Of The Review:**

This is a good paper overall. The method provides an improvement over an existing method that makes is more applicable in real-world settings where OOD data is common. The text is well written and experiments are clearly laid out. Empirical results are mixed, in some settings performing on-par with existing methods and in some improving on them.

---

> ### Author Response · Authors · 2022-11-11
> **Response to Reviewer R1Ae**
>
> Dear Reviewer R1Ae,
>
> Thank you for your valuable feedback and comments.
>
> ---
>
> **Would trivial collapse be a problem in RoPAWS?**
>
> RoPAWS also favors the non-collapsing property of representations. Recall that the proof logic of Proposition 1 in the PAWS paper is as follows:
> 1. When the representation collapses, the distances between representations $d(z,z_i)$ become identical for all $z_i$ in the labeled batch $\mathcal{B}_l$.
> 2. Then, the prediction $p$ is given by the average of class labels $y_i$, which becomes a uniform distribution due to the class-balanced sampling.
> 3. However, the target sharpening makes the target $p^+$ not equal to the uniform distribution; hence, $\nabla_\theta H(p^+, p) > 0$, and the model escapes from the representation collapse.
>
> RoPAWS follows the same logic: the prediction $p$ becomes a uniform distribution when representation collapses, being escaped by step 3 above. The difference from PAWS is that now the prediction is the average of true labels $y_i$ of labeled batch $\mathcal{B}_l$ and pseudo-labels $\hat{y}_i$ of unlabeled batch $\mathcal{B}_u$. Here, the initial belief of pseudo-labels $\hat{y}_i$ becomes a uniform distribution by averaging the true labels $y_i$, and averaging this uniform distribution keeps the updated belief uniform after label propagation. We added the discussion in Appendix L of our revised manuscript.
>
> ---
>
> **Improvements on curated data**
>
> RoPAWS improves PAWS for curated data since it refines the pseudo-labels via label propagation. This benefit is more significant when the original pseudo-label of PAWS is inaccurate, e.g., only a few labels are available. As a result, RoPAWS gives a large gain (+2.3%) for CIFAR-10 using 25 labels per class. However, the gap saturates when using more labels, e.g., 250 labels per class for CIFAR-10 or 10K (1%) or 100K (10%) labels for ImageNet. The table below shows the test accuracy of models trained on curated CIFAR-10 using 4 labels per class. We report the average and standard deviation over five runs. RoPAWS highly improves (+3.0%) PAWS in this more challenging scenario.
>
> \begin{array}{ccc}
> \text{FixMatch} & \text{PAWS} & \text{RoPAWS (ours)} \newline
> \hline
> 86.2\pm{3.4} & 85.7\pm{2.4} & 88.7\pm{2.2}
> \end{array}
>
> We provide additional analyses verifying the benefit of RoPAWS. Specifically, RoPAWS **reduces confirmation bias** [1] and **improves calibration** [2] of pseudo-labels. Recall that lowering wrong predictions (i.e., less confirmation bias) and understanding right confidence (i.e., better calibration) play a crucial role in label propagation.
>
> First, we verify that label propagation refines the pseudo-labels and reduces confirmation bias (i.e., improves correctness). To this end, we present the accuracy of pseudo-labels over label propagation, following the same setup of Appendix H. The table below shows the test accuracy of RoPAWS trained on curated CIFAR-10 using 25 labels per class, using different epochs of checkpoints (total: 600) and different soft nearest-neighbor (Soft-NN) classifiers. Iter. 0 means the Soft-NN classifier only uses labeled data, and Iter. $k \ge 1$ means the iterations of label propagation (using both labeled and unlabeled data) where $k = \infty$ is the fixed-point solution. The pseudo-label becomes more accurate as iteration goes on. Also, note that the effect of label refinement is more impactful in early epochs, when the pseudo-labels are inaccurate. This correction makes RoPAWS converge to better optima.
>
> \begin{array}{lccccc}
> \text{Epoch} & \text{Iter. 0} & \text{Iter. 1} & \text{Iter. 2} & \text{Iter. 3} & \text{Iter. }\infty \newline
> \hline
> 10 & 73.8 & 74.6 & 74.6 & 75.0 & 75.0 \newline
> 100 & 87.9 & 88.7 & 88.7 & 88.7 & 88.7 \newline
> 600 & 95.0 & 95.3 & 95.3 & 95.3 & 95.3 \newline
> \end{array}
>
> Second, we compare the calibration of PAWS and RoPAWS classifiers. To this end, we measure expected calibration error (ECE) [3]. We apply the TorchMetric implementation [4] and use the default 15 bins. The table below shows the ECE (lower is better) of models trained on curated CIFAR-10 using 25 labels per class. RoPAWS gives better calibration than PAWS.
>
> \begin{array}{cc}
> \text{PAWS} & \text{RoPAWS (ours)} \newline
> \hline
> 1.52 & 0.92 \newline
> \end{array}
>
> We added these additional results and discussion in Appendix K of our revised manuscript.
>
> [1] Arazo et al. Pseudo-Labeling and Confirmation Bias in Deep Semi-Supervised Learning. IJCNN 2020.\
> [2] Loh et al. On the Importance of Calibration in Semi-supervised Learning. arXiv 2022.\
> [3] Guo et al. On Calibration of Modern Neural Networks. ICML 2017.\
> [4] https://torchmetrics.readthedocs.io/en/v0.10.2/classification/calibration_error.html
>
> ---
>
> Sincerely,\
> Authors

---

### Official Review · Reviewer_yYAe · 2022-10-29

**Confidence:** 3
**Correctness:** 3
**Technical Novelty And Significance:** 2
**Empirical Novelty And Significance:** 2
**Recommendation:** 5

**Clarity, Quality, Novelty And Reproducibility:**

[Clarity]
The current manuscript is no problem in terms of clarity.

[Quality]
I am unsure which kind of quality should be discussed here; however, the current manuscript does not have sufficient quality for acceptance due to the lack of novelty justifications against the previous methods.

[Novelty]
As presented in the Weakness section, the current manuscript fails to justify the novelty of the proposed method against the previous method SSKDE.

[Reproduciblity]
I think that it is no problem for reproducibility.

**Strength And Weaknesses:**

[Strength]

S1. Representation learning is one of the hot topics at this conference. Technically solid methods for this topic will draw attention from a broad range of researchers and engineers.

S2. The current manuscript is basically well-written and easy to follow.

[Weakness]

W1. I am afraid that the main idea of the proposed method is similar to semi-supervised kernel density estimation (SSKDE) presented in the following paper.
Wang, Hua, Mei, Hong, Qi, Song, Dai, "Semi-supervised kernel density estimation for video annotation," Computer Vision and Image Understanding, Volume 113, Issue 3, 2009. https://doi.org/10.1016/j.cviu.2008.08.003.
I understand that the specific implementation of the proposed method is different from that of the above paper; however, the advantages of the proposed method against SSKDE should be clearly presented both literally and experimentally.

**Summary Of The Paper:**

This paper deals with the problem of semi-supervised representation learning and proposes a method based on the specific previous method called PAWS. More specifically, the previous method PAWS can be regarded as kernel density estimation (KDE) for modeling the distribution of latent variables, and the proposed method called RoPAWS extends it in a semi-supervised way. The proposed method also introduces data-dependent in-domain priors assuming that an example is likely to have the same label if it is close to some labeled example.

**Summary Of The Review:**

I have to recommend this paper be rejected since it requires major revisions to justify the novelty of the proposed method. It should conduct detailed bibliographical surveys and experimental comparisons with previous related methods.

---

> ### Author Response · Authors · 2022-11-11
> **Response to Reviewer yYAe**
>
> Dear Reviewer yYAe,
>
> Thank you for your valuable feedback and comments.
>
> ---
>
> **Comparison with SSKDE**
>
> Thank you for suggesting the related work, SSKDE [1]. As you pointed out, both RoPAWS and SSKDE compute the posterior probability of labels based on the KDE over the feature space. However, SSKDE has several critical limitations:
> - SSKDE requires an ad-hoc feature extractor. SSKDE applies KDE to the precomputed features, unlike RoPAWS jointly learning the representation and the KDE generative classifier. The ability to learn good representations is one of the main differences between RoPAWS and pre-deep-learning approaches like SSKDE.
> - SSKDE is infeasible for large-scale data. Since SSKDE is a nonparametric method, it computes the kernel matrix between all training data. Thus, the training cost of SSKDE is $O(N^3)$ (or $O(N^2)$ with approximation), where $N$ is the number of training data. In contrast, RoPAWS only computes the kernel matrix within the mini-batches, reducing the training cost to $O(N * M^3)$, where M is the size of the mini-batch and $M \ll N$. This enables the training under arbitrarily large data.
>
> Due to the limitations listed above, SSKDE cannot be applied to real-world large-scale datasets without significant method modification.
>
> In addition, RoPAWS has multiple technical novelties over SSKDE:
> - The target problem is different. The main goal of RoPAWS is robust semi-supervised learning under uncurated data. Thus, we propose in-domain prior to handling out-of-distribution (OOD) samples. In contrast, SSKDE aims for the curated setup and does not address the overconfidence issue of OOD samples. In addition, SSKDE mainly targets the video annotation problem and only conducts small-scale experiments, while RoPAWS verifies its effectiveness on realistic large-scale image benchmarks.
> - The definition of kernel is different. RoPAWS discovers that PAWS is implicitly modeling KDE with cosine similarity kernels, where the kernel definition comes from the design of recent self-supervised learning methods. In contrast, SSKDE uses Gaussian or Exponential kernels, which may not fit with deep learning techniques.
> - The formulation and required assumption are different. RoPAWS leverages the class-balanced sampling of PAWS to simplify the formula, canceling out the denominator of $p(x|y)$ in the Bayes rule. SSKDE considers a different assumption that $p(y|x)$ is given by a linear combination of pseudo-labels $\hat{p}(y|x)$ and ground-truth labels (if exists), where $\hat{p}(y|x)$ is again a function of $p(y|x)$ and kernels. As a result, the final prediction formula of SSKDE in Eq. (14) of their paper and RoPAWS in Eq. (11) are different.
>
> Despite the fundamental limitations of SSKDE, we provide an additional experiment comparing RoPAWS and SSKDE for the readers’ interest. Since SSKDE requires an ad-hoc feature extractor, we run SSKDE upon PAWS and RoPAWS backbones, in addition to a randomly initialized linear layer of dimension 3x32x32$\to$128. We follow the default configuration of SSKDE: use the Gaussian kernel with gamma = 1/n_features [2], weight for the linear interpolation $t = 0.9$, and number of iterations $M = 40$.
>
> The table below shows the test accuracy of models on CIFAR-10 using 25 labels per class. First, one can see that the quality of the feature extractor mostly decides the final accuracy. We remark that RoPAWS learns a good representation, which is a clear advantage over SSKDE. Second, the Soft-NN classifier (default of PAWS/RoPAWS) performs better than the SSKDE classifier. Indeed, the representation of PAWS/RoPAWS is optimized for the Soft-NN classifier. The results confirm that RoPAWS is empirically better than SSKDE. However, exploring different designs of generative classifiers (e.g., extending SSKDE to learn representations jointly) would be an interesting future research direction.
>
> \begin{array}{l|c|cc|cc}
> \text{Feature extractor} & \text{Random linear} & \text{PAWS} & \text{PAWS} & \text{RoPAWS} & \text{RoPAWS} \newline
> \text{Inference} & \text{SSKDE} & \text{SSKDE} & \text{PAWS (Soft-NN)} & \text{SSKDE} & \text{RoPAWS (Soft-NN)} \newline
> \hline
> \text{Test acc.} & 13.9 & 91.0 & 92.4 & 92.8 & 94.7 \newline
> \end{array}
>
> We added the discussion and comparison with SSKDE in the Related Work section and Appendix J of our revised manuscript.
>
> [1] Wang et al. Semi-supervised kernel density estimation for video annotation. CVIU 2009.\
> [2] https://scikit-learn.org/stable/modules/generated/sklearn.metrics.pairwise.rbf_kernel.html
>
> ---
>
> Sincerely,\
> Authors

---

### Author Response · Authors · 2022-12-05
**A Gentle Reminder**

Dear reviewers and AC,

We deeply appreciate your time and efforts in reviewing our manuscript.

As reviewers highlighted, our work tackles an important problem (yYAe,Q9GX,NjZV,oxEy) with a technically solid method (yYAe,R1Ae,oxEy), validated with thoughtful experiments (R1Ae,Q9GX,oxEy) presented in a well-written manuscript (All).

In our response, we addressed all the concerns and questions of the reviewers. In particular, we resolved the concerns of negative reviewers, verifying that RoPAWS outperforms prior works under various scenarios:
* (yYAe) Comparison with prior work, SSKDE
* (NjZV) Additional comparison with prior robust SSL methods

In addition, we provided experiments to answer the questions of positive reviewers:
* (R1Ae) Can RoPAWS escape the representation collapse?
* (R1Ae) Why does RoPAWS also improve curated data?
* (oxEy) How does the choice of data augmentation impact RoPAWS?
* (oxEy) How do the self-supervisory signals of labeled data impact RoPAWS?

The feedback from the reviewers enhanced our work to clearly demonstrate the technical novelty and empirical gain over prior works and strengthened the analysis.

We believe that we sincerely and successfully addressed all the concerns with supporting experimental results. Please consider checking the response and adjusting your assessment.

Thank you very much!

Best regards,\
Authors

---

### Decision · Program_Chairs · 2023-01-20

**Decision:**

Accept: poster

**Justification For Why Not Higher Score:**

The current paper mainly extends PAWS into RoPAWS, with several natural but somewhat incremental improvements, such KDE for density-based calibration, in-domain prior. The final training objective in Eq. 14 is closely fit into the PAWS loss with modifications in the instance weights and calibrated predictions. Empirical results show that these modifications lead to significant performance increases on some de facto datasets such as CIFAR, but the benefit is minor in ImageNet. Hence, both the technical contributions and the empirical benefits are above the acceptance threshold but do not reach that of the spotlight.

**Justification For Why Not Lower Score:**

The contribution warrants acceptance as a poster. Also the review scores support such a decision. It is not possible to reject a paper with scores (88855) because most of the concerns were addressed while the AC cannot find further flaws.

**Metareview: Summary, Strengths And Weaknesses:**

This paper presents RoPAWS, a robust semi-supervised representation learning method for both curated and uncurated data, e.g. out-of-distribution data. As the coined name implies, the approach is largely built upon the previous approach PAWS (Assran et al., 2021), but introduces several necessary improvements based on the reinterpretation of PAWS as a generative classifier that models densities using kernel density estimation (KDE). RoPAWS is shown to significantly outperform PAWS and other leading SSL methods on several common datasets.

The paper has received five review reports with mixed opinions. During the revision and rebuttal phase, authors addressed most of the concerns on technical novelty and empirical evaluation. There are still two votes of "Weak reject" after the discussion phase. Nonetheless, one of the negative reviewer said in the discussion thread that (s)he would increase the score (although not done so finally).

AC read the paper as well as all the reviewing materials, and concluded that the most concerns in the "Weak reject" reviews, such as the technical connection to SSKDE, empirical evaluation with more methods on more datasets, have been addressed reasonably well. As a final note, the paper improves PAWS with a new generative interpretation and several modifications which are not technically surprising but empirically beneficial for real-world uncurated data analysis. The presentation of the paper is good, with sufficient supplementary material such as implementation details and source code for release. Therefore, AC recommends to accept the paper in its revised form as a poster.

**Note From Pc:**

if the above contains the word "oral" or "spotlight" please see: "oral" presentation means -> notable-top-5% and "spotlight" means -> notable-top-25%. As stated in our emails, we are disassociating presentation type from AC recommendations